# Age-Dependent Decline in Neuron Growth Potential and Mitochondria Functions in Cortical Neurons

**DOI:** 10.3390/cells10071625

**Published:** 2021-06-29

**Authors:** Theresa C. Sutherland, Arthur Sefiani, Darijana Horvat, Taylor E. Huntington, Yuanjiu Lei, A. Phillip West, Cédric G. Geoffroy

**Affiliations:** 1Department of Neuroscience and Experimental Therapeutics, Texas A&M Health Science Center, Bryan, TX 77807, USA; tcsutherland@tamu.edu (T.C.S.); sefiani@tamu.edu (A.S.); d-horvat@tamu.edu (D.H.); taylore.hunt94@tamu.edu (T.E.H.); 2Department of Microbial Pathogenesis and Immunology, Texas A&M Health Science Center, Bryan, TX 77807, USA; abbylei@tamu.edu (Y.L.); awest@tamu.edu (A.P.W.)

**Keywords:** mitochondria, aging, neurite growth, dysfunction, mitochondrial respiration, CNS Injury

## Abstract

The age of incidence of spinal cord injury (SCI) and the average age of people living with SCI is continuously increasing. However, SCI is extensively modeled in young adult animals, hampering translation of research to clinical applications. While there has been significant progress in manipulating axon growth after injury, the impact of aging is still unknown. Mitochondria are essential to successful neurite and axon growth, while aging is associated with a decline in mitochondrial functions. Using isolation and culture of adult cortical neurons, we analyzed mitochondrial changes in 2-, 6-, 12- and 18-month-old mice. We observed reduced neurite growth in older neurons. Older neurons also showed dysfunctional respiration, reduced membrane potential, and altered mitochondrial membrane transport proteins; however, mitochondrial DNA (mtDNA) abundance and cellular ATP were increased. Taken together, these data suggest that dysfunctional mitochondria in older neurons may be associated with the age-dependent reduction in neurite growth. Both normal aging and traumatic injury are associated with mitochondrial dysfunction, posing a challenge for an aging SCI population as the two elements can combine to worsen injury outcomes. The results of this study highlight this as an area of great interest in CNS trauma.

## 1. Introduction

Spinal Cord Injury (SCI) is the second most common cause of paralysis and results in varying degrees of motor and sensory dysfunction over the lifetime of the patient. The international incidence rate of SCI is approximately 13 cases per 100,000 population [1]. The incidence in the United States is approximately 54 cases per million population, with 17,730 new cases each year [2,3]. The demographic of the SCI population has seen an important shift in recent decades. The average age at injury is increasing, as is the age of the SCI population. Currently, the average age of incidence in the US is 43 years old [3]. There are two peaks in incidence, one in young adults (20–30 years old) and one in the aging population (≥65 years old) adults [1]. As a result of the increasing incidence of SCI in middle-aged and aging populations and the fact that people with SCI are living longer, the average age of people who reported paralysis due to an SCI is now approximately 48 years old [4]. There has been limited consideration of the effects of aging on SCI in current SCI research. In contrast to the human SCI population, the majority of preclinical research is performed in young adult rodents (2–4 months), and less than 0.35% of experimental rodents used are 12 months or older (representing 40 years of age in humans) [5]. This is likely to be a significant impediment to translating preclinical research into viable clinical therapies.

An age-dependent decline in axon growth has been reported in a variety of model organisms, both vertebrates and invertebrates. This decline is influenced by both neuron-intrinsic [6] and extraneuronal [7] factors. Previous studies have shown that even genetic manipulation of neuron-intrinsic factors aimed at promoting axon growth is sensitive to age, decreasing efficiency and efficacy at various ages [8]. This observation suggests that there are other significant factors at play in the age-dependent decline in axon growth potential. One intrinsic factor of particular interest is neuronal mitochondria. The mitochondrial theory, or free-radical theory, is a prominent theory of aging. It identifies mitochondrial decline as a fundamental mechanism of biological and cellular aging [9,10,11,12]. This theory has been proposed for decades with reactive oxygen species (ROS) at its core. However, in recent years, evidence has shown that other facets of mitochondrial dysfunction are important in the process of cellular aging. Extensive work implicates mitochondria as a central figure, but there are many questions still to be answered [11]. Mitochondria are involved in a plethora of vital cellular functions including energy production, intracellular homeostasis, calcium balance, and the metabolism of various dietary substrates, and immunity [9]. There is extensive evidence that the functional decline of mitochondria contributes to normal aging throughout the body, playing a prominent role in age-related processes such as cellular senescence, chronic inflammation, and decreased stem cell activity [12]. The decline in mitochondrial function has been associated with a variety of diseases, including age-related neurodegeneration [9]. Mitochondrial dysfunction is considered one of the hallmarks of the aging brain [13]. 

The most prominent role of mitochondria is cellular respiration and energy production, which is significant in SCI because neuron and axon growth requires large amounts of cellular energy. Any dysfunction in oxidative phosphorylation (OXPHOS) and the electron transport chain (ETC) may have a significant impact on aging neurons’ ability to support axon sprouting and growth. Additionally, these organelles are also involved in calcium (Ca^2+^) homeostasis and buffering, which is vital to neuronal function. Mitochondria at the synapse are involved in buffering Ca^2+^ to regulate neurotransmission [14]. Previously, it has been observed that Ca^2+^ buffering is impaired in the aging brain [15]. The mitochondrial permeability transition pore (mPTP) becomes more active with increased ROS and oxidative stress. The prolonged opening of this channel affects both ROS balance and Ca^2+^ homeostasis. Other mitochondrial membrane channels, such as the translocase of the outer membrane (TOM) [16,17] and the inner membrane (TIM) [18,19] complexes, are essential to the translocation of a variety of bioactive proteins and molecules into the mitochondria [20,21,22]. OXPHOS Complex I subunit protein GRIM19 [23,24] plays a role in maintaining mitochondrial membrane potential (ΔΨM) [23] and in transport across the membrane [20]. Alteration in the expression and function of these proteins and complexes, as well as decreases in membrane potential, affect mitochondria’s ability to effectively import essential molecules and maintain cellular Ca^2+^ levels. 

Mitochondria play important roles in normal aging [12], axon growth [25,26,27,28,29,30], and the progression of SCI [31,32,33]. Significantly, mitochondrial dysfunctions and detrimental oxidative stress are associated with both normal aging and central nervous system (CNS) trauma. This poses a challenge for an aging SCI population; these two factors may compound each other and worsen the outcome for aging SCI patients. Using a new system of adult cortical neuron isolation and culture, we compared neuronal mitochondrial function in mice in four age groups: 2 months (young), 6 months (young adult), 12 months (middle-aged), and 18 months (aging). We observed a variety of detrimental changes in neuronal mitochondria in the aging CNS that will have a significant effect on both neuronal health and the ability to promote neurite growth after injury. Collectively, our results suggest an important role for mitochondrial functions in the age-dependent decline in axon growth potential. This also may suggest the exploration of promoting neuronal mitochondrial function to promote axon growth in injured neurons regardless of age or time postinjury as a promising future direction. These observations need to be expanded in vivo and strategies to manipulate the mitochondria need to be analyzed in detail. 

## 2. Materials and Methods

*Animals*: This study used female wild-type C57Bl/6 mice of 4 age groups; 2 months, 6 months, 12 months, and 18 months. All procedures were conducted according to Institutional approved Animal Use Protocols. 

*Isolation of Adult Cortical Neurons*: Adult cortical neurons were isolated in an enriched population from the cortex using the Mitlenyi MACS system, the Adult Brain Dissociation Kit (130-107-677), and Adult Neuron Isolation Kit (130-126-603). The mice were euthanized using CO^2^, and the brains were removed. The cortex was isolated using blunt dissection under a microscope and transferred to a MACS C-tube. Samples were dissociated with two mice per tube. This was then dissociated using the appropriate pre-set protocol on the Miltenyi gentleMACS Octo Dissociator with heaters. The manufacturer’s recommended protocols were followed for both the Adult Brain Dissociation Kit and Adult Neuron Isolation Kit (using LS columns) with minor adjustments. During this dissociation and isolation protocol, the neurons are stripped of their neurites and axons, allowing this to be a model for axonal injury and neurite growth in vitro. This protocol was used as the basis for all of the subsequent analyses performed in this study.

*RT-qPCR for Isolated Cell Population Purity:* RNA was extracted from both the negative fraction (neurons) and positive fraction using Directzol RNA micro-prep columns (Zymo, Irvine, CA, USA; R2061) directly following MACS dissociation and isolation (D0), as well as after 2- and 4-days in vitro and Seahorse analysis. RNA concentration was measured using NanoDrop (Thermo Fisher, Waltham, MA, USA), and samples were calculated accordingly. cDNA was synthesized using Quantabio cDNA Synthesis Kit (Quanta, Gaithersburg, MD, USA; 95047), followed by qPCR with Quantabio Perfecta Sybr Fastmix + Rox (Quanta, 95073) run on ViiA7 Real-Time PCR System (Life Technologies, Grand Island, NY, USA), as previously published [34]. Primers were used to identify the main cellular constituents of the isolated populations: Neuron- MAP2 (F: 5′-CTG GAG GTG GTA ATG TGA AGA TTG; R: 5′-TCT CAG CCC CGT GAT CTA CC-3′) and NeuN (F: 5′-AAC CAG CAA CTC CAC CCT TC-3′; R: 5′-CGA ATT GCC CGA ACA TTT GC-3′); Astrocytes- GFAP (F: 5′-CTA ACG ACT ATC GCC GCC AA-3′; R: 5′-CAG GAA TGG TGA TGC GGT TT-3′) and Glast (F; 5′-CAA CGA AAC ACT TCT GGG CG-3′; R: 5′-CCA GAG GCG CAT ACC ACA TT-3′); and Oligodendrocytes- Oligo2 (F; 5′-GAA CCC CGA AAG GTG TGG AT-3′; R: 5′-TTC CGA ATG TGA ATT AGA TTT GAG G-3′); as well as β-actin (F: 5′-CTC TGG CTC CTA GCA CCA TGA AGA-3′; R: 5′-GTA AAA CGC AGC TCA GTA ACA GTC CG-3′) for normalization. This analysis was performed in triplicate on available young neuron samples on DO (2–6 month) (N = 2), and in young (2–6 month) (N = 2) and aging (12–18 month) (N = 2) samples collected after seahorse analysis. The neuron enrichment in the negative fraction was calculated as −ΔCT of NeuN against GFAP and Glast using the formula: −ΔCT = −(ΔCT NeuN − (SQRT(ΔCT GFAP^2^ + ΔCT Glast^2^))). 

*Analysis of Neurite Growth In Vitro:* The enriched cortical neuron population resulting from MACS dissociation and isolation of female mice was plated onto poly-D-lysine (PDL) and laminin-coated #1.5 thickness glass culture plates. At the time of plating, these cells were without axons and significant neurites due to the dissociation process. These cells were cultured in MACS Neuro media (Miltenyi, Auburn, CA, USA) with penicillin (50 units/mL)/streptomycin (50 μg/mL) (Corning, Corning, NY, USA), L-Alanyl-L-glutamine (Sigma-Aldrich, St. Louis, MO, USA; 2 mM), B-27 Plus Supplement (Gibco, Grand Island, NY, USA, 1×), and brain-derived neurotrophic factor (BDNF; Tonbo Biosciences, San Diego, CA, USA, 1 μg/mL), with 50% media replacement every second day, and fixed in 8% paraformaldehyde (PFA; 15 min) at the end of 7 days. Immunocytochemistry was performed on PFA fixed cells for βIII-Tubulin (1:500, BioLegend) with a green fluorescent secondary antibody (AF488; 1:500, BioLegend, San Diego, CA, USA) and cells were imaged using the ImageXpress automated confocal system (Molecular Devices, Ismaning, Germany) using 10x objective (3 samples/age). Neurite growth was analyzed from the 10x βIII-Tubulin images using NeuronJ (ImageJ plugin; NIH, Bethesda, MD, USA) to measure average neurite length/cell, total neurite length/cell, the number and length of primary (1°), secondary (2°), and tertiary (3°) neurites, and the number of branching points visible (N = 50–75 cells/age, in 3 wells). Cells were counted as neurons and measured only if they were βIII-Tubulin positive and the neurites were ≥10 µm.

*qPCR of Mitochondrial DNA:* After MACS isolation (as described previously) the enriched cortical neuron cells were immediately disrupted and lysed for RNA and DNA extraction using an AllPrep DNA/RNA Micro Kit (Qiagen, Hilden, Germany, 80284). The protocol was performed to the manufacturer’s recommendation for this kit. Mitochondrial DNA analysis by qPCR was performed on the extracted DNA, as previously published [35]. Briefly, DNA was subjected to qPCR using PerfeCTa SYBR Green FastMix (Quanta) and the CFX384 Real-Time System (BIO-RAD, Hercules, CA, USA). Each sample analyzed consisted of the complete cortex from 2 mice pooled and was run in triplicate normalized against nuclear-encoded 18 s using the 2^−∆∆Ct^ method. Relative expression (%) analysis was performed as described using primers specific to multi-copy cytosolic ribosomal DNA sequence 18 s (F: 5′-CTT AGA GGG ACA AGC GGC G-3′; R: 5′-ACG CTG AGC CAG TCA GTG TA-3′) and mitochondrial ribosomal DNA sequence 16 s (F: 5′-GTT ACC CTA GGG ATA ACA GCG C-3′; R: 5′-GAT CCA ACA TCG AGG TCG TAA ACC-3′). Only two genes were used here due to the small amount of DNA in each sample. These were analyzed as 2-, 6-, 12-, and 18-month samples (N = 2/age, in triplicate), and as grouped young (2- and 6-month samples) and older (12- and 18-month samples) (N = 4/group, in triplicate).

*Analysis of Mitochondrial Respiration*: Cortical neurons were isolated from female mice as described above, with each sample consisting of the cortices from 2 mice pooled together. Isolated cortical neurons were plated at 40,000 cells/well onto a 96-well Seahorse XFe96 plates treated with PDL and laminin and left to adhere and begin neurite sprouting over 4 days. After 4 days in vitro, these cells were analyzed on the Agilent Seahorse XFe Analyzer using the Mito Stress Test Kit. This kit uses three drug cocktails added into the assay media over time to alter the cell respiration—oligomycin (2.5 µM), trifluoromethoxy carbonylcyanide phenylhydrazone (FCCP; (2.0 µM), and rotenone/antimycin A (0.5 µM). The oxygen consumption rate (OCR) is measured to assess mitochondrial respiration, and the extracellular acidification rate (ECAR) and proton efflux rate to assess glycolysis. This analysis was run to the manufacturer’s suggested protocol. The experiment was repeated twice independently, with similar results, and therefore, the results from both were merged to increase the number of wells analyzed. Due to the small cell yield and low baseline OCR in the 12- and 18-month neurons, the wells were grouped into two age ranges. The 2- and 6-month neurons, and the 12- and 18-month neurons from both plates were grouped into “young” and “older”(N = 7/group), respectively, to increase the power of the analysis. OCR graphs and functional mitochondrial measures were calculated using the Agilent Seahorse report generator macro for the Mito Stress Test. This macro uses the raw OCR and ECAR data collected from the experiment at baseline and after drug application in Wave (Agilent software, La Jolla, CA, USA) and calculates basal and maximal respiration, ATP production, proton leak, nonmitochondrial oxygen consumption, spare respiratory capacity, and coupling efficiency. 

*Western Blot for OXPHOS Protein Complexes:* After isolation of an enriched cortical neuron population (from 2 brains/sample, female) using the previously described protocol, the isolated cell populations were immediately placed in 20 µL RIPA protein extraction buffer (EMD Millipore, Merck Group, Darmstadt, Germany) with HALT protease/phosphatase inhibitors (ThermoFisher Scientific, Waltham, MA, USA) and PMSF (Cell Signalling Technology, Danvers, MA, USA). Protein extraction was performed on ice for 30 min with intermittent mixing. The protein supernatant was separated by centrifuging at 3000× *g* for 30 min and stored at −80 until use. Sample concentrations were measured using NanoDrop (Thermo Fisher) and 50 μg of protein was run for each sample (N = 3/age). Protein was run on Novex 10% Tris-Glycine Mini Gels (Thermo Fisher) using Novex Tris-Glycine SDS Running Buffer (Thermo Fisher) and NuPAGE Sample Reducing Agent (Thermo Fisher). Gels were run for 2 h and 10 min at 80 V and transferred using the Trans-Blot SD Semi-Dry Transfer Cell system (Bio-Rad). After transfer, the LI-COR Revert 700 Total Protein Stain (LI-COR, Lincoln, NE, USA) was applied, per manufacturer’s protocol, and imaged using LI-COR Odyssey CLx Imaging System (LI-COR). Following this, the membranes were treated using SuperSignal Western Blot Enhancer (Thermo Fisher) for 10 min before blocking in 5% Milk in TBST for 1 h. After washing in 0.05% TBST, Total OXPHOS Rodent Antibody Cocktail (Abcam, Cambridge, UK; 1:2000) was added to the membranes and incubated overnight at room temperature. After washing off the primary antibody, the secondary antibody (Anti-Mouse IgG HRP-linked; Cell Signaling, Danvers, MA, USA; 1:2000) was added and incubated for 1 h, followed by Supersignal West Femto Maximum Sensitivity Chemiluminescent Substrate (Thermo Fisher) for 5 min. The samples were analyzed 6 times in repeated Western blots. The blots were imaged using the LI-COR Odyssey Fc Imaging System (LI-COR Biosciences), chemiluminescent setting (an example shown in Appendix A). Protein expression, as measured by band intensity (mean greyscale value; MGV), was normalized to the total protein stain. 

*Functional Assay for Mitochondrial ATP Production:* Neurons were isolated from female mice, as described above with MACS. Cells were manually counted under a microscope using 0.4% Trypan Blue (VWR, Radnor, PA, USA, 97063-702) and inserted into a Levy Counting Chamber (Hausser Scientific™ 3900, Horsham, PA, USA) to determine cell number in each sample. Each sample consisted of the cortical neurons preparation from 2 mice pooled (N = 3 samples/age) and each sample was run in triplicate as described below. The enriched cortical neuron population was immediately placed in 50 µL RIPA protein extraction buffer (EMD Millipore, Merck Group, Darmstadt, Germany) with HALT protease/phosphatase inhibitors (ThermoFisherScientific) and PMSF (Cell Signalling Technology) to lyse the cells and mitigate ATPase activity. After a 30 min incubation at 4 °C with intermittent mixing, the supernatant was separated by centrifuging at 20,000× *g* for 20 min. Immediately following lysing, the relative ATP concentration of isolated neurons was measured using the bioluminescence-based ATP Determination Kit (Invitrogen, Carlsbad, CA, USA, A22066) following the manufacturer’s recommended protocol. Briefly, 6 µL of each sample, standard, or control was mixed with 60 µL of Standard Reaction Buffer (1 mM dithiothreitol, 0.5 mM D-luciferin, 1.25 µg/mL firefly luciferase, Component E Reaction Buffer) before being loaded onto a 384-well polystyrene µCLEAR^®^ bottom plate (Greiner Bio-One GmbH, Frickenhausen, Germany, 781091) and incubated at 28 °C for 10 min. Relative Light Units (RLU) were measured at 28 °C using the Synergy 2 SL Microplate Reader with Gen5™ software (BioTek, Winooski, VT, USA). 

Known ATP standards were run alongside samples to determine ATP concentration. The ATP concentration of lysed from 2-, 6-, 12-, and 18-month-old cortical neurons was determined using a linear equation developed from the ATP standards: Concentration of ATP (nM) = 0.0035 × Relative Light Units (RLU) (R^2^ = 0.9753). Using the volume of the assay, the number of cells lysed, and the ATP concentration of each sample, the approximate number of ATP molecules in cortical neurons from each age group was determined. The number of ATP/cell was calculated and expressed as a percentage change relative to the youngest group in the analysis (2 months, or 2–6 months), this measure is shown in the resulting graphs. 

*Flow Cytometry Analysis of Membrane Potential:* JC-1 (5,5,6,6′-tetrachloro-1,1′,3,3′ tetraethylbenzimi-dazoylcarbocyanine iodide) is a lipophilic cationic dye frequently used to analyze mitochondrial membrane potential (ΔΨM). JC-1 has normally green fluorescence until it forms into aggregates, which have red fluorescence. In healthy cells with a normal ΔΨM, JC-1 enters and accumulates in the mitochondria at high levels and forms red fluorescent (PE) aggregates. In unhealthy or stressed cells, whose mitochondria have increased membrane permeability and loss of electrochemical potential, JC-1 enters the mitochondria at decreased levels that are not sufficient for the formation of aggregates maintaining its normal green fluorescence (FITC) [36].

Cortical Neurons were isolated from female mice using the MACS protocol previously described, with one mouse/sample (N = 8/age). Samples were isolated and analyzed two at a time—young versus aged, 2- against 12-month-old samples, and 6- against 18-month-old samples. Following isolation, the samples were split in half; one half was treated with FCCP at 40 µM for 5 min as a control, before JC-1 was added to all the cell samples, and incubated in Neuron media (37 °C, 5% CO_2_) for 20 min. After this incubation, additional media was added to these samples and they were centrifuged. The cell pellet was suspended in FACS buffer (PBS, 5% FBS, 1% BSA, 0.05% Sodium Azide) and analyzed using a BD Fortessa flow cytometer with BD Diva software using (using PE, FITC, and BV421 lasers). Sytox Blue dead cell stain was added to the samples immediately before analysis. Overall, 50,000 events were analyzed per sample. These samples were first gated using forward (FCS) and side scatters (SSC), followed by height/area for single cells, and then live cells based on Sytox staining. From this population of live, single cells the duel-positive population (both PE and FITC) was analyzed for PE and FITC fluorescent intensity (MFI). This was expressed and analyzed as a red:green ratio (PE/FITC).

*Mitochondrial Membrane-Associated Protein Immunocytochemistry:* The cortical neurons plated and cultured for 7 days that were used for neurite analysis were also stained for either TOM20 (1:200, Proteintech, Wuhan, China), TIM23 (1:200, Santa Cruz Biotech, Santa Cruz, CA, USA) or GRIM19 (1:200, Santa Cruz Biotech) with a red fluorescent secondary antibody (AF555; 1:500). Cells were imaged using the 20× objective on a Zeiss Axio Observer system. The mean greyscale value (MGV) per area of the individual neuron was analyzed using ImageJ (N = 50–80/age). Specifically, individual neurons, including the cell soma and processes, were manually outlined in ImageJ, and the area and MGV of the AF555 fluorescence were measured for each cell. This was used to calculate the MGV/area of each cell as a measure of fluorescent intensity of the antibody highlighted in the AF555 channel (either TOM20, TIM23, or GRIM19).

*Data Analysis:* All quantifications and measurements were performed by experiments blinded to the age groups. Seahorse data were analyzed in Agilent Wave and Seahorse Analytics XF software. Flow cytometry data were analyzed using FlowJo (v10.7.1, LLC, Ashland, OR, USA). Normally distributed data were analyzed using ANOVA with Tukey’s post hoc test, and unpaired t-tests using Graphpad Prism. All histograms are representative of the data mean with error bars showing the standard error of the mean (SEM). 

## 3. Results

### 3.1. Cell Population Purity of Enriched Cortical Neurons after Isolation and Culture

RT-qPCR was used to analyze the cellular constituents of the enriched cortical neuron population. Two neuron-associated genes, two astrocyte-associated genes, and an oligodendrocyte-associated gene were analyzed, normalized against internal β-actin. Neuron enrichment was calculated as −ΔCT = −(ΔCT NeuN − (SQRT(ΔCT GFAP^2^ + ΔCT Glast^2^))). RNA extracted from the neuron enriched negative fraction immediately following MACS dissociation and isolation (D0) showed a neuron enrichment −ΔCT of 10.123. After 4 days in vitro with neuron-specific culture methods, and analysis using the Agilent Seahorse, the neuron-enriched population had a −ΔCT of 11.051. This corroborates preliminary culture data (unpublished) in which we observed 81.5 ± 8.2% βIII-Tubulin positive neurons and <2 ± 1.5% cells expressing GFAP, with no microglia or oligodendrocytes observed. Olig2 expression was not observed in samples from 2-day and 4-day in vitro. Therefore, we are confident the data presented below are from a highly enriched neuronal population.

### 3.2. Neurite Growth Potential Declines in Mature Cortical Neurons In Vitro with Age

Analysis of in vitro cortical neuron neurite outgrowth cultured for 7DIV from 2-, 6-, 12- and 18-month-old mice showed decreases with age as well as altered patterns of growth (Figure 1). The total length of neurite growth per cell decreased with age, though this was only significant (*p* < 0.05) between the 2-month and both the 12- and 18-month neurons (Figure 1A). The significance of this decrease is magnified (*p* < 0.0005) when comparing grouped young (2- and 6-month-old) and older (12-and 18-month-old) neurons (Figure 1Ai). The average neurite length also significantly decreases in older neurons, compared to young both between the four separate ages (Figure 1B) and in grouped ages (Figure 1Bi). The number of primary neurites was similar in young and older neurons (Figure 1C,Ci); however, the length of the longest primary neurite decreased significantly between young and older neurons (Figure 1D,Di). Conversely, the number of secondary neurites was increased in older neurons compared to young (*p* < 0.05), and the number of branching points (between primary, secondary, and tertiary neurites) is also significantly increased in older neurons (Figure 1E,Ei). This suggests a decrease in neurite growth potential with age and increased branching in aged neurons, compared to their younger counterparts (as displayed in Figure 1F–Gi). 

### 3.3. Expression of Mitochondrial DNA and Mitochondrial DNA Copy Number Changes with Age

Analysis of mitochondrial copy number was performed using quantitative PCR of 18 s (nuclear) and 16 s (mitochondrial) genes. Due to the small sample size from isolated cortical neuron populations, only one nuclear and one mitochondrial gene was used. Relative expression of the mitochondrial 16 s ribosomal DNA sequence normalized to the nuclear-encoded 18 s ribosomal DNA sequence was calculated using the 2-month samples as a baseline. Separately, both the 12- and 18-month neurons had significantly higher relative expression (%) than the 6-month neurons (*p* < 0.005), and the 18-month neurons were also significantly increased, compared to the 2-month neurons (*p* < 0.05) (Figure 2A). When grouped into young (2 and 6 months old) and older (12–18 months old), this relative expression was significantly increased in older neurons, compared to their younger counterparts (*t*-test; *p* = 0.0004) (Figure 2B). This indicated an increase in mtDNA with age.

### 3.4. Mitochondrial Respiration of Cultured Mature Cortical Neurons Changes with Age

The Mito Stress Test (Agilent) measures both OCR and ECAR at baseline levels and in response to drugs modulating mitochondria activity. First, oligomycin acts on complex V to inhibit the conversion of ADP to ATP; then FCCP alters membrane potential and stimulates maximal capacity of the respiratory chain; and finally, rotenone inhibits complex I, and antimycin A inhibits complex III to suppress the ETC function (Figure 3A). This results in a response curve as seen in the 2- and 6-month neurons (Figure 3B). Interestingly, we did not observe this strong response in the 12- and 18-month neurons, with only very subtle increases and decreases in OCR at the response time points (Figure 3B). The baseline OCR for the 12- and 18-month neurons was clearly lower than that seen in the 2- and 6-month neurons (Figure 3B); however, there was no significance in the majority of parameters. From the OCR measurements, we observed a trend for decreased basal respiration (Figure 3C), maximal respiration (Figure 3D), ATP production (Figure 3E), and proton leak (Figure 3F) in the 12- and 18-month group, compared to the 2- and 6-month group. ECAR is associated with lactate efflux, and increases in this measure indicate cellular activation and proliferation. The ECAR measure showed a decrease in older neurons, compared to young neurons. Together, these measures suggest increased efficiency and respiratory capacity in younger cortical neurons, while older neurons are more quiescent and unresponsive. 

### 3.5. Expression of Mitochondrial OXPHOS Complexes in the Electron Transport Chain Is Consistent in Cortical Neurons of Different Ages

Expression of the five OXPHOS complexes of the ETC (C I-C V) was analyzed using Western blot (Appendix A). Comparison of 2-, 6-, 12- and 18-month neuronal protein extract (directly after brain dissociation and cell separation) showed no significant alterations in the expression of any of the five complexes (Figure 4A). When grouped into young (2- and 6-months old) and older (12- and 18-months old), no significant changes in any of the five complexes were observed (Figure 4B). Across all age groups, complex I and complex V showed the highest expression. There was no statistical significance in these findings using either ANOVA or *t*-tests.

### 3.6. Intracellular ATP Increases in Neurons with Age

Mitochondrial ATP production was measured using a bioluminescence-based ATP assay and compared between different age cohorts. Comparison of the number of intracellular ATP molecules in 2-, 6-, 12-, and 18-month-old neurons showed a trend of increase with age, which was signed between the 2- and 12-month neurons (Figure 5A; *p* < 0.05). When grouped into young (2- and 6-months old) and older (12- and 18-months old) neurons, there was a significant increase in the relative number of intracellular ATP molecules (Figure 5B; *p* < 0.05). There was an upward trend in the number of intracellular ATP molecules with age, and a simple linear regression equation was calculated to predict the number of intracellular ATP molecules based on the age of the cohort (*p* = 0.1302, R^2^ = 0.7566) (Figure 5C). 

### 3.7. The Mitochondrial Membrane Potential of Isolated Cortical Neurons Changes with Age

We next assessed the change in mitochondrial membrane potential (ΔΨM) in freshly isolated cortical neurons of different ages. We used JC-1 flow cytometry to measure the PE:FITC mean fluorescent intensity (MFI) ratio in cortical neurons. Our data showed that ΔΨM was altered in an age-dependent manner. Indeed, the PE:FITC MFI ratio declined with age; however, this was not significant in either grouped data (*p* = 0.09) or between the individual ages, due to high standard deviations in the data (Figure 6). This age-dependent alteration may have significant implications for mitochondrial function in cortical neurons. 

### 3.8. Mitochondrial Membrane-Associated Proteins Are Altered in Aging Neurons

Immunocytochemistry for several mitochondrial membrane transport proteins showed a change in their expression with age (Figure 7). TOM20 is a prominent translocase of the outer mitochondrial membrane [16,17] and was observed to decrease with age between the young (2- and 6-months old) neurons and their older counterparts (12- and 18-months old) (Figure 7A,Ai). It was also noted that the TOM20 immuno-staining appeared less visible in the long neurites of the older neurons, though the cell bodies were just as bright as the young (Figure 8). TIM23 is a translocase of the inner membrane that shuttles molecules into the mitochondrial matrix from the intermembrane space [18,19]. TIM23 expression was surprisingly increased in the older neurons, compared to the younger ones. Both 12- and 18-month neurons showed significantly increased TIM23 MGV/cell, compared to the 2-month neurons (*p* < 0.05 and *p* < 0.0001, respectively). Moreover, 18-month neurons were significantly higher in TIM23 than 6-month neurons (*p* < 0.005) (Figure 7B). When grouped into young (2- and 6-months old) and older (12 and 18 months old), the significance between young and old increased (*p* < 0.0001), with the older neurons showing nearly double the TIM23 MGV/cell (Figure 7Bi). GRIM19 is a subunit of complex I of the ETC [23,24] that is involved in transport across the membrane [20] and maintenance of membrane potential [23]. GRIM19 expression showed a trend to decrease with age, though this was not significant except between the 6- and 18-month neurons (Figure 7C). When combined into young (2- and 6-month old) and older (12- and 18-month old) groups, there was a significant decrease in GRIM19 expression (*p* < 0.05) seen with age (Figure 7Ci).

## 4. Discussion

The lack of functional recovery associated with SCI is linked to the degeneration of severed axons, the abnormal growth of what remains, and the failure of axons to regrow. Axonal growth inhibitors are also present after SCI, associated with elements of the ongoing secondary injury [7]. Consequently, axon growth has received a lot of attention in SCI research. Studies have investigated the mechanisms behind it, how it is affected by injury, and most significantly, how growth and regeneration can be promoted after injury. Despite rapid progress in understanding and manipulating axon growth postinjury, the impact of age on axon growth and regeneration in the CNS has not been characterized. Mitochondria are vital to axon growth and regeneration [29,30]. These organelles are known to decrease in efficacy and functionality with age and after SCI, representing a point where aging and SCI intersect.

This study represents the first comparison of primary cortical neurons isolated from mice at different ages. Using a modified dissociation and cell culture system, we were able to culture cortical neurons from mice up to 33 months old (not shown). This allows for a deep understanding and characterization of the different neuron-intrinsic factors that mediate the observed decline in growth and recovery potential seen in aging [6]. We observed a decrease in neurite length and an increase in branching in older cortical neurons. This clearly demonstrates the age-dependent decline in cortical neurons neurite outgrowth in vitro, suggesting that neuron-intrinsic properties are involved in this reduction. The protocol used in this study did not allow for the identification of the neurites as axons or dendrites. Further specific staining is needed to explicitly demonstrate the age-dependent decline in axon growth in cortical neurons. While only the intrinsic properties of the cortical neurons were assessed in this in vitro model, we acknowledge that these neurons are also impacted by a wide range of extraneuronal factors in vivo [7] and that both neuron-intrinsic and extrinsic factors have to be taken into account to fully comprehend the complexity of the different factors reducing axonal and dendritic growth in the CNS. This work is also the first to specifically address changes in mitochondrial functions with age in cortical neurons. Overall, the age-related mitochondrial changes observed (increase in mtDNA and ATP production, reduction in respiration and mitochondrial membrane potential, increase in TIM23, and reduction of TOM20 and Grim19) support the hypothesis that mitochondrial functions in cortical neurons are reduced with age. This could, in part, explain the age-dependent decline in growth potential. However, we also recognize that a direct causative effect remains to be demonstrated.

Mitochondrial DNA (mtDNA) is a multi-copy genome located in the mitochondrial matrix that encodes for 13 proteins of the ETC. An intact and functional mitochondrial genome is required for normal mitochondrial function. Damage and/or depletion of mtDNA have important effects on the expression and function of the OXPHOS proteins of the ETC and energy production. The subsequent mitochondrial malfunctions have been implicated in a wide range of disorders including aging and neurodegeneration [37]. In this study, the multi-copy cytosolic ribosomal DNA sequence 18s and mitochondrial ribosomal DNA sequence 16s were used to estimate the ratio of mitochondrial to nuclear genomes and therefore the number of mitochondrial to each cell. Surprisingly, this analysis showed increased relative expression of mtDNA in middle-aged and aging mice, compared to younger adults. This could have multiple implications for cortical neurons. 

The increase in mtDNA may be a compensatory mechanism to promote ETC protein expression, ATP production, and cell survival in response to the dysfunctional state of neuronal mitochondria with age. Previous work in the lungs has shown an increase in mtDNA copy number in elderly individuals. This increase was linked to oxidative stress responses and suggested as a possible mechanism to compensate for defects in mitochondria retaining mutated mtDNA [38]. Other previous studies have consistently reported an age-related decline in mtDNA copy number (mtDNAcn) [39,40,41,42], but these results are confounded by mixed cell samples and contamination from platelets, which inflate mtDNAcn and decline in number with age [43]. A study specifically examining peripheral blood mononuclear cells observed a positive correlation between mtDNAcn and age [43]. This may be used by cells as a compensatory upregulation to counteract the loss of functionally intact mitochondria with age [44,45]. While the increase in mtDNA expression was paralleled by increased ATP levels, the expression levels of OXPHOS complexes were not increased with age, as would be expected from an increase in mtDNA. A potential explanation is that the mtDNA transcription is not as efficient in older neurons, and more mtDNA is needed to produce a similar level of OXPHOS proteins and function. This has previously been observed in the aging heart [46]. Alternatively, the OXPHOS proteins half-life may be reduced in old cortical neurons, and a quick turnover is necessary to maintain the OXPHOS protein level. These possibilities need to be explored further in older neurons.

The increase in mtDNA could also indicate a reduction in mitophagy in older neurons that results in increased numbers of dysfunctional or declining mitochondria retained in the cells. While increased mtDNA is generally associated with an increased number of mitochondria in a cell, it is also possible that the reduction of autophagy with age artificially increases the mtDNA levels. However, the mtDNA is dysfunctional and poorly transcribed. Following this contention, an increase of mtDNA in older neurons might compensate for this reduction in transcriptional activity and lead to similar levels of OXPHOS proteins. A previous study in the brain and spinal cord has demonstrated an increase in mtDNA damage with age, associated with multiple dysfunctions, reductions in NADH/FADH mediated respiration, and detrimental increases in ROS [47]. However, this study examined whole CNS tissue, as opposed to the cortical neuron cell population examined in the current paper. There have been several previous studies that have shown an increase in mtDNAcn but not necessarily a corresponding increase in mitochondrial function, both in disease [48,49,50] and aging [43]. A recent study in polymerase gamma (POLG) mutator mice showed elevated mtDNA copy number but decreased OXPHOS function, despite having normal OXPHOS complex levels [49].

A final possibility is that the amount of mtDNA is associated with the location of the mitochondria in the cell. During the dissociation process, axons and dendrites are sheared from the cell body; thus, the mtDNA extracted immediately from these isolated neurons is mostly from the cell soma. The increase in mtDNA in older neurons may correlate with the clustering of the mitochondria in the soma and decreased transport into the processes. Indeed, TOM20 staining appeared to be less prominent in the neurites of older cortical neurons in culture, which may support this idea. Previous papers have reported increased mitochondrial clustering around the nucleus when the cells are inclined to enter apoptosis [51,52,53,54]. This may also be linked, as aging cells are more stressed and vulnerable than their younger counterparts. 

The primary function of mitochondria is cellular respiration and energy production via the generation of ATP through OXPHOS. In the normal functioning CNS, cortical neurons rely on significant ATP production to maintain cellular functions [55,56]. The formation of active axonal growth cones is required for neurons to promote axonal growth. This is essential for regeneration and restoration of function after injury and requires even more substantial energy production [28]. Decreases in either the expression or function of OXPHOS complexes in the aging CNS may have a significant impact on mitochondrial bioenergetics and, subsequently, the ability to produce sufficient energy to support growth. The observed age-related decline in mitochondrial activity [57,58,59] poses a challenge in SCI as successful axon regeneration from cortical neurons requires an increase in mitochondrial dynamics [26,27,60]. A significant decline in mitochondrial function is observed as early as 12 h after SCI and progressively worsens with time [31]. Therefore, the loss of mitochondrial bioenergetics with age may compound SCI.

Mitochondria have reduced bioenergetics and declining function with age [9,11,61]. This led us to anticipate a decrease in the expression of some, if not all, of the five OXPHOS complexes with age. However, we did not observe this decrease in cortical neurons. This was surprising as other measures examined suggested dysfunctional mitochondrial respiration. Using the Agilent Seahorse system to measure dynamic respiratory functions, we observed the reduction of basal respiration and ATP production in older neurons, although nonstatistically significant. These measures indicate that the young neurons are healthier cells that are more able to produce the energy required for neurite growth in vitro conditions. The older neurons, as well as having decreased respiratory measures, were less responsive to the drugs used in the Mito Stress Test. This supports the idea that these cells are less metabolically active and less able to respond to cellular stressors. However, these older isolated neurons measured at the low end of the desirable baseline OCR range for this assay, which calls for further exploration to strengthen these observations. A previous study using the Seahorse system to measure OCR from mitochondria isolated from different brain regions also demonstrated a reduction in OCR with age in rats. However, the alterations to respiratory complexes with age demonstrated in this previous paper were region specific and combined the mitochondria from all cell populations present [62]. Taken together, the results from the OXPHOS Western blot and the dynamic metabolic measurements from the Seahorse suggest that there is not a decrease in expression of OXPHOS complexes in older cortical neuron mitochondria but instead a dysfunction of respiration and energy metabolism in the existing ETC. Indeed, it is possible that the assembly of respiratory complexes or supercomplexes is impaired [63]. Further investigation of each OXPHOS complex and their enzymatic activity in cortical neurons (as previously analyzed in isolated mitochondria from whole brain tissue [62]) would be helpful to answer the questions raised by these results.

A decline in energy production, energy expenditure, and basal metabolic rate with age has been documented for decades [64,65,66]. The number of ATP molecules present in cortical neurons is therefore expected to decline with age. Our results suggest an increase in ATP from young adult (2 and 6 months old) to middle-aged (12 months old) and aged (18 months old) mice. There may be several explanations for this surprising observation. First, the number of ATP molecules present is not directly connotative of the metabolic rate, expenditure, or efficiency of an organism. Aging can reduce ATP/AMP ratio [67], suggesting reduced protein efficiency, concentration, or kinetics. Neurons in aged mice may have trouble utilizing free ATP, associated with decreases in their metabolic rate. Direct analysis of ATPase activity is required to elucidate age-related changes in neuronal energy metabolism. 

Secondly, it is possible that the elevated ATP levels recorded in aging cortical neurons, as well as the increased mtDNA expression, could be related to a compensatory upregulation of glycolysis [67] and perhaps mitochondrial biogenesis to support the declining respiratory function. The lower membrane potential observed with age, as well as the reduced respiratory measurements and decreased ECAR, are consistent with this idea. Thirdly, the assay used measures ATP present in the cell at the time of lysis rather than dynamic ATP production. Therefore, the absolute ATP numbers may be higher in older mice due to a dysfunctional or slower metabolism of ATP. This could indicate increased storage but under-utilization of the ATP produced. Alternatively, for various reasons, older animals may require more ATP to perform than their younger counterparts [68]. It must also be taken into account that this ATP measurement and the seemingly contradictory data from the Seahorse respiratory analysis are produced from two different experiments, under two different conditions. Specifically, this static ATP measure is from cell lysate analysis while the Seahorse measurements are taken from live cells in vitro, and we do not know how this may affect the ATP levels. 

Finally, this observation may be linked to euthanasia. The length of CO_2_ exposure may impact the amount of free ATP. The anaerobic environment of a CO_2_ chamber may induce rapid utilization of free ATP molecules in young cortical neurons to prevent death. Hypoxia shifts ATP production towards glycolytic metabolism [69]. As aging increases glycolytic ATP production [67], the anaerobic environment of the CO_2_ chamber may be more favorable to older neurons and therefore result in artificially high ATP levels. These data support the contention that aging reduces the ability of cells and systems to react to changes in metabolic demand and the ability to utilize stored energy. However, a more in-depth examination is required to elucidate the processes underlying the age-dependent increase in ATP and to clarify the mechanisms of ATP storage and utilization in older neurons and their biological implications for neuron survival and axon growth. 

This study found a decrease in the mitochondrial membrane potential (ΔΨM) in middle-aged and older cortical neurons, compared to younger neurons. The decreased PE prevalence in the JC-1 stained neurons suggests an increase in membrane depolarization in older neurons. ΔΨM is involved in mitochondrial respiratory rate, ATP synthesis, Ca^2+^ uptake and storage, and ROS balance. It is an important factor that can be used to identify dysfunctional mitochondria [70,71]. ΔΨM is involved in energy storage in the mitochondria and, alongside the proton gradient, creates the transmembrane hydrogen ion gradient needed to produce ATP. ΔΨM is necessary for the import of cations across the membrane into the mitochondrial matrix and the export of anions [70]. Incomplete coupling of mitochondria results in a natural proton leak across the inner membrane and some dissipation of energy. We observed a decreased proton leak in older neurons. Previously, aging mitochondria have shown reduced ΔΨM and an increased rate of proton conductance [72]. The decreased proton leak in older neurons could be indicative of compromised energy homeostasis. It may also be linked to the observed ΔΨM depolarization and be another indication of dysfunctional respiration. 

Mitochondria in aged neurons also have Ca^2+^ buffering deficits as well as reduced ΔΨM [73,74]. In previous studies, mitochondria extracted from whole brain tissue have shown region-specific increases in ROS production and impaired Ca^2+^ buffering with age [15]. Opening of the mPTP can induce ΔΨM depolarization, loss of oxidative phosphorylation, and release of apoptotic factors, and may be a precipitating event in apoptosis. Conversely, ΔΨM depolarization may be a consequence of the apoptotic-signaling pathway rather than an early precipitating factor [75]. In either case, loss of ΔΨM can be considered an indicator of dysfunctional mitochondria and increased sensitivity to apoptotic insult. This is in keeping with the results of this study and the suggested dysfunction and increased vulnerability of cortical neurons with age. 

Membrane transport is vital to moving molecules into and out of the mitochondrial matrix and therefore is linked to the ETC, the ΔΨM, proton gradient, and ROS homeostasis [76,77]. Two prominent families of multipart membrane complexes are the translocase of the outer membrane (TOM) [16,17] and the inner membrane (TIM) [18,19]. TOM20 is responsible for the recognition and translocation of mitochondrial preproteins synthesized in the cytosol. We observed a trend for reduced TOM20 expression in older cortical neurons, suggesting a potential decrease or dysfunctional entry of proteins into the mitochondria. Some studies have reported no change in TOM protein in aging skeletal muscle [78,79], while other studies using mitochondrial fractionation have suggested upregulation of the TOM proteins in aging skeletal muscle [80]. The increase in TOM protein was particularly observed in older subjects that were functionally inactive [78], suggesting an upregulation in mitochondrial protein import may be a compensatory mechanism for other age-related dysfunction in skeletal muscle mitochondria with age. While we observed a decrease in TOM20 expression with age in cortical neurons, there was a significant increase in TIM23. This may be linked to a similar compensatory mechanism to try to boost dysfunctional aging mitochondria. TOM complex is the primary point of entry for most proteins directed to the mitochondrial matrix [81], and TIM 23 is the primary translocase for matrix proteins to move through the inner mitochondrial membrane [18]. TIM23 is known to interact with subunits of the ETC and can play an important role in inserting inner-membrane proteins when the ΔΨM is reduced [54]. Overexpression of TIM23 in Parkinson’s disease has been observed to reduce neurodegeneration. The neuroprotective mechanism of TIM23 increase may be linked to an increased rate of protein import into mitochondria and also stabilization of complex I subunits [54]. Mitochondrial membrane transport, and especially TIM23, has been linked to neurodegenerative disorders, with data suggesting that deficient protein import is an early event in the development of Huntington’s disease [82]. In this study, TIM23 knockdown in primary cortical and striatal neurons resulted in increased neuronal death in a Huntington’s disease mouse model [82]. We observed a significant increase in the expression of TIM23 in cortical neurons with age, which may indicate a compensatory mechanism to help promote mitochondrial function and increase survival.

Gene associated with retinoid interferon-induced cell mortality 19 (GRIM19) was first identified as an apoptotic nuclear protein associated with tumor cells [24]. To date, there have been multiple roles identified for GRIM19, including as a subunit to mitochondrial ETC complex I [21,24] and as a transport protein that helps to shuttle specific molecules to act in the mitochondria [20,22,83]. Significantly, in mitochondria, GRIM-19 is required for electron transfer activity of complex I, and disruption of ΔΨM in GRIM-19 mutant models enhances sensitivity to apoptotic stimuli [23]. We observed a trend of decreased GRIM19 expression with age in cortical neurons. This decrease may be linked to the depolarized ΔΨM and dysfunctional respiration also observed. We observed no reduction in expression of OXPHOS complex I with aging. However, the decrease in GRIM19 may support a decline in function of the OXPHOS system rather than a decline in expression of the ETC proteins. 

Past studies have compared mitochondria isolated from the brain tissue of animals to observe a plethora of age-related alterations [13]. In vitro studies using neurons [84] or astrocytes [85] from brains of both young and older mice, or cells “aged’’ in culture, have suggested that most cell types in the brain likely experience the accumulation of dysfunctional mitochondria during aging. This observation is borne out by the results of the current study that suggest an increase in mitochondrial number (as suggested by mtDNA expression) in aging but dysfunctional indicators across the range of mitochondrial processes examined. It is likely that the decline in functional mitochondria, not in mitochondrial number, is linked to the less efficient neurite growth observed. The retention of dysfunctional mitochondria seen in aging may also result in an increased number of mitochondria present while having a detrimental impact on neuron health and neurite outgrowth. 

After CNS injury, the mitochondria translocate into the injured axons, increasing the mitochondria density [25]. A failure to increase mitochondrial function in neurons is linked to poor regeneration [25]. Another possibility linked to decreased growth potential in older neurons is that the mitochondria in older neurons are not being transported to where they are needed to support axon growth. A recent study using a syntaphilin knockout mouse model, which lacks the static anchor protein to hold axonal mitochondria stationary, hence resulting in increased axonal mitochondrial motility, showed improvements in three distinct models of SCI and axonal injury [86]. These results suggest that enhancing mitochondrial transport and motility can enhance recovery after SCI in young mice [86]. However, this contention has yet to be tested in an aging paradigm. Some observations from the current study (increase in mtDNA from direct isolation and membrane protein expression more prominent in the soma) may hint at a reduction in transport in older neurons that needs to be explored. The regulatory mechanisms that transport, distribute, and clear mitochondria in neurons are compromised in neurotrauma [87], which may compound the retention of dysfunctional mitochondria that is seen in aging. Mitochondrial energetics are beneficial to axon regeneration in SCI in young mice [86]; however, this has not been examined in middle-aged and aging animals. The differences in mitochondrial function and dynamics in older neurons may present a challenge, but they also offer a target that can be used to improve SCI recovery in aging patients.

## 5. Conclusions

Mitochondria are vital to the growth and regeneration of axons after injury, and this capability decreases in the aging CNS. The mitochondrial alterations we have observed in older neurons may play a significant role in the age-dependent decline in axon growth potential. In an aging SCI population, this may have important implications. Age and injury may compound one another and result in worse outcomes for patients. The wide range of alterations and the magnitude of them do not point to one particular factor but instead to a general decline in function affecting neuronal mitochondria with age. This bears greater scrutiny to allow targeting of mitochondria in aging neurons with the eventual goal of pursuing a potentially efficacious therapy for SCI in patients of all ages.

## Figures and Tables

**Figure 1 cells-10-01625-f001:**
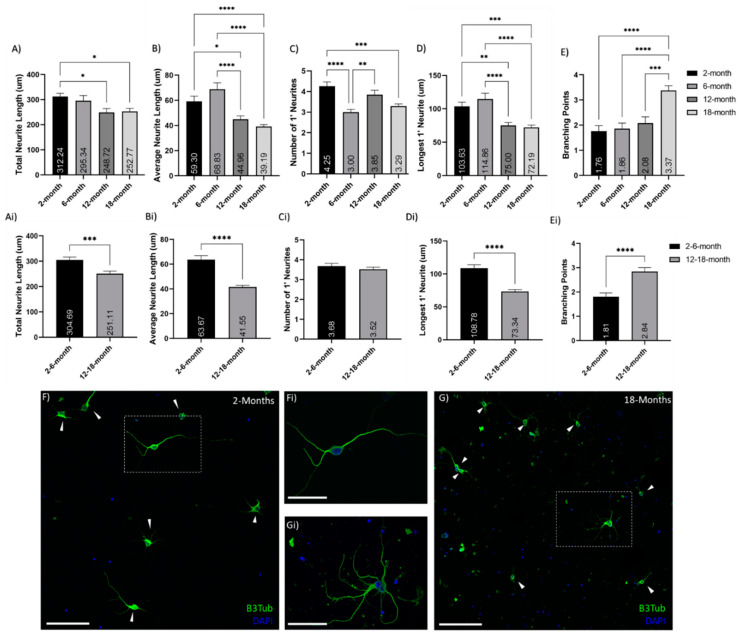
Reduction of neurite outgrowth with age. Histograms of neurite analysis and feature quantifications comparing between (**A**–**E**) 2-, 6-, 12- and 18-month-old neurons, as well as (**A**–**E**) grouped young (2- and 6-month-old) and older (12- and 18-month-old) neurons. These show decreased neurite growth and increases branching in older neurons, compared to young: (**A**,**Ai**) show age-related decreases in total neurite growth per cell; (**B**,**Bi**) show decreases in the average neurite length with age; (**C**,**Ci**) show the number of primary (1′) neurites per cell; (**D**,**Di**) show a decline in primary neurite length with age; (**E**,**Ei**) indicate an increase in neurite branching with age. This pattern is illustrated in the representative 20× confocol images of (**F**) 2-month and (**G**) 18-month cortical neuron cultures after 7 days in vitro; stained with βIII-Tubulin (Green) and DAPI (Blue); inset (**Fi**,**Gi**) show 60× magnification of neurons. These images also show the increased debris and less healthy look of the old neurons after 7 DIV. * (*p* < 0.05), ** (*p* < 0.005), *** (*p* < 0.0005), **** (*p* < 0.0001). N = 50–75 cells/age. Graphs show mean and SEM. Arrows indicate individual neurons. Scale bars are 100 μm for F and G; 50 μm for Fi and Gi.

**Figure 2 cells-10-01625-f002:**
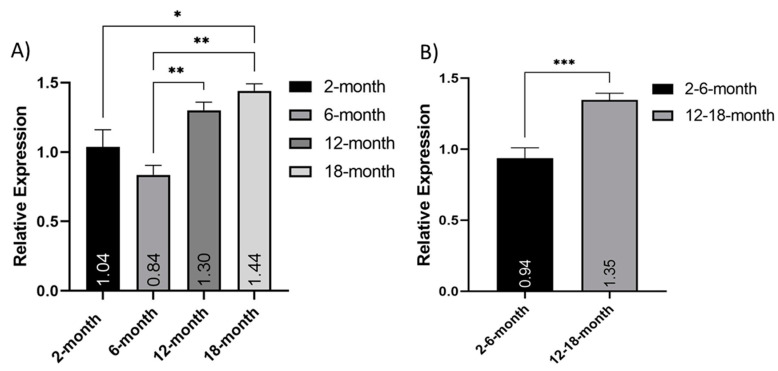
Increase in Mitochondrial DNA expression with age. Relative expression of the mitochondrial *16s* ribosomal DNA sequence normalized to the nuclear-encoded 18 s ribosomal DNA sequence: (**A**) comparing 2-, 6-, 12- and 18-month-old isolated neuron populations (*N* = 2); (**B**) comparing grouped young (2- and 6-months old) and older (12- and 18-months old) (N = 4). * (*p* < 0.05), ** (*p* < 0.005), *** (*p* < 0.0005). Graphs show mean and SEM.

**Figure 3 cells-10-01625-f003:**
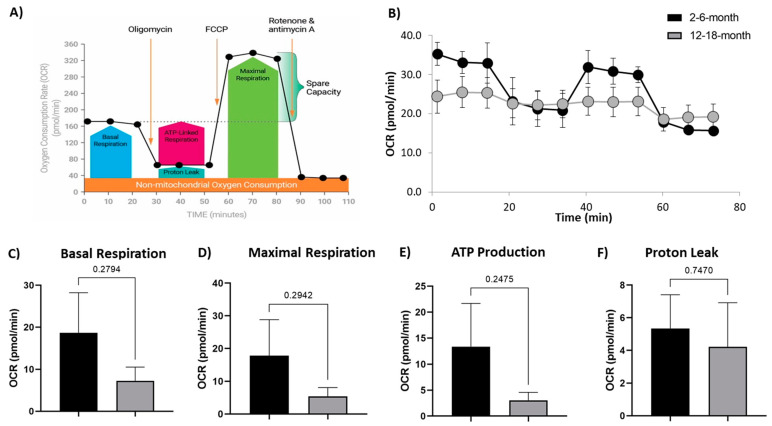
Alteration to Mitochondrial respiration in older neurons: (**A**) schematic of the injected drugs and time course of the Seahorse Mito Stress Test (Agilent) showing an example timecourse of the OCR response to the administered drugs, and the respiratory measures calculated; (**B**) oxygen consumption rate (OCR) over the time course of the assay shows the expected response curve and higher OCR in the young (2- and 6-month) neurons, compared to lower OCR and almost no responsiveness to the mitochondrial active drugs in the grouped 12- and 18-month neurons (older); (**C**) basal respiration; (**D**) maximal respiration; (**E**) ATP production; (**F**) proton leakage were all decreased in older cortical neurons, compared to the young. There was no statistical significance to these trends. N = 7/age group. Graphs show mean and SEM.

**Figure 4 cells-10-01625-f004:**
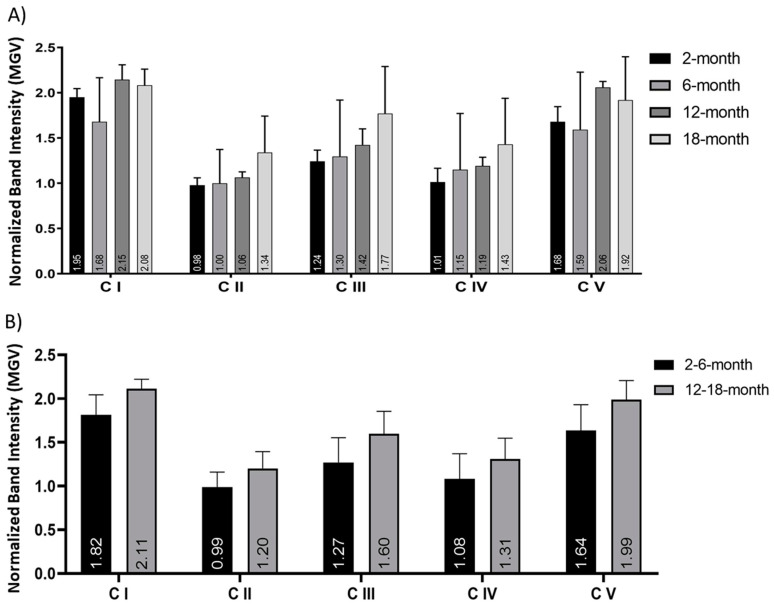
No significant change in the Mitochondrial OXPHOS Complex protein expression in cortical neurons. Histograms of the Western blot band intensity, calculated as mean greyscale value (MGV) and normalized to the total protein stain, of proteins of all five OXPHOS complexes (C I-C V) in the ETC: (**A**) comparison of 2-, 6-, 12- and 18-month cortical neurons and (**B**) grouped young (2- and 6-months old) versus aging (12- and 18-months old) neurons indicated no significant change in expression with age. This was not statistically significant in any comparison. N = 3 samples/age, repeated 6 times. Histograms show mean and SEM.

**Figure 5 cells-10-01625-f005:**
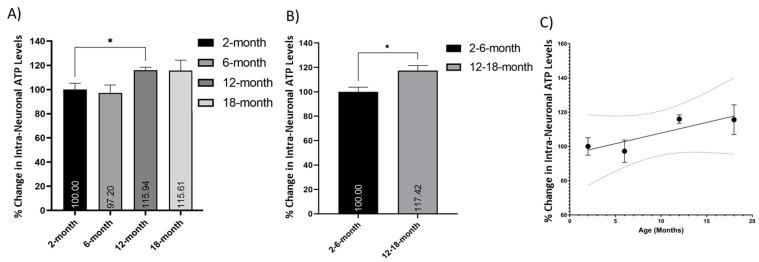
Increased retention of ATP molecules in older neurons. Histograms of the number of intracellular ATP molecules per cell, as measured using a bioluminescence-based ATP assay, expressed as a percentage change relative to the youngest group (young or 2 months old, respectively): (**A**) comparison of 2-, 6-, 12- and 18-month cortical neurons, and (**B**) young (2- and 6-months old) versus older (12- and 18-months old) neurons indicated a trend of increase in intracellular ATP with age; (**C**) this trend of increases can be expressed as a simple, nonsignificant linear relationship (*p* = 0.1302, R^2^ = 0.7566) with 95% CI (dotted line). * (*p* < 0.05). N = 3 samples/age, analysed in triplicate. Histograms show mean and SEM.

**Figure 6 cells-10-01625-f006:**
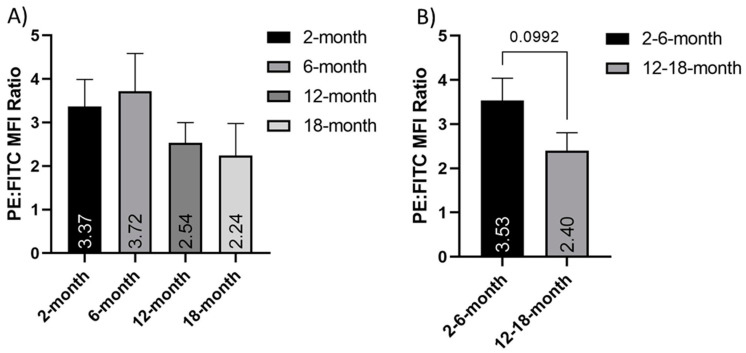
Depolarization of the mitochondrial membrane in older neurons. Histograms of the PE:FITC mean fluorescent intensity (MFI) ratio of JC-1 stained cortical neurons normalized to FCCP treated controls, both in (**A**) 2-, 6-, 12- and 18-month-old isolated neuron populations and (**B**) grouped young (2- and 6-months old) and older (12- and 18-months old) neurons. These show a nonsignificant age-dependent decrease in mitochondrial membrane potential. N = 8. Graphs show mean and SEM.

**Figure 7 cells-10-01625-f007:**
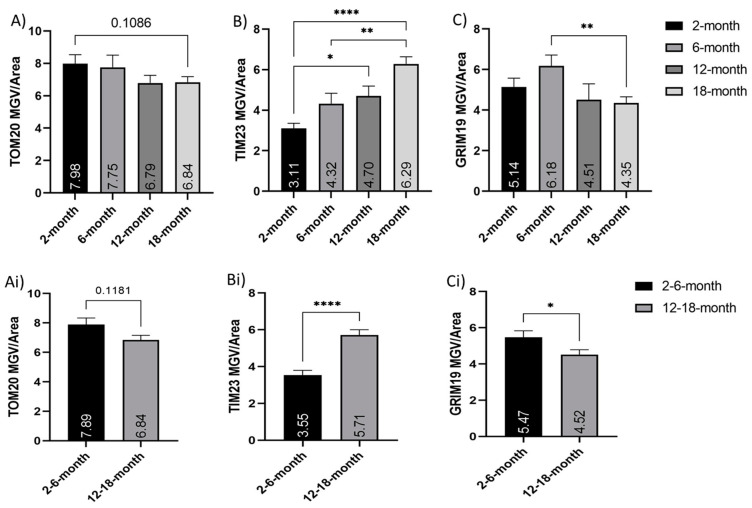
Alterations in expression of membrane transport-associated proteins with age. Histograms of TOM20, TIM23, and GRIM19 fluorescent mean grayscale value/cell (MGV/cell): (**A**–**C**) comparison of young (2- and 6-months old) and older (12- and 18-months old) neurons, as well as (**A**–**C**) between 2-, 6-, 12- and 18-month=old neurons; (**A**,**Ai**) show an age-related trend for a decrease in TOM20 expression per cell, though these were not significant; (**B**,**Bi**) show significant increases in the TIM23 expression with age; (**C**,**Ci**) show decreases in GRIM19 MGV per cell. * (*p* < 0.05), ** (*p* < 0.005), **** (*p* < 0.0001). N = 50–75 cells/age. Graphs show mean and SEM.

**Figure 8 cells-10-01625-f008:**
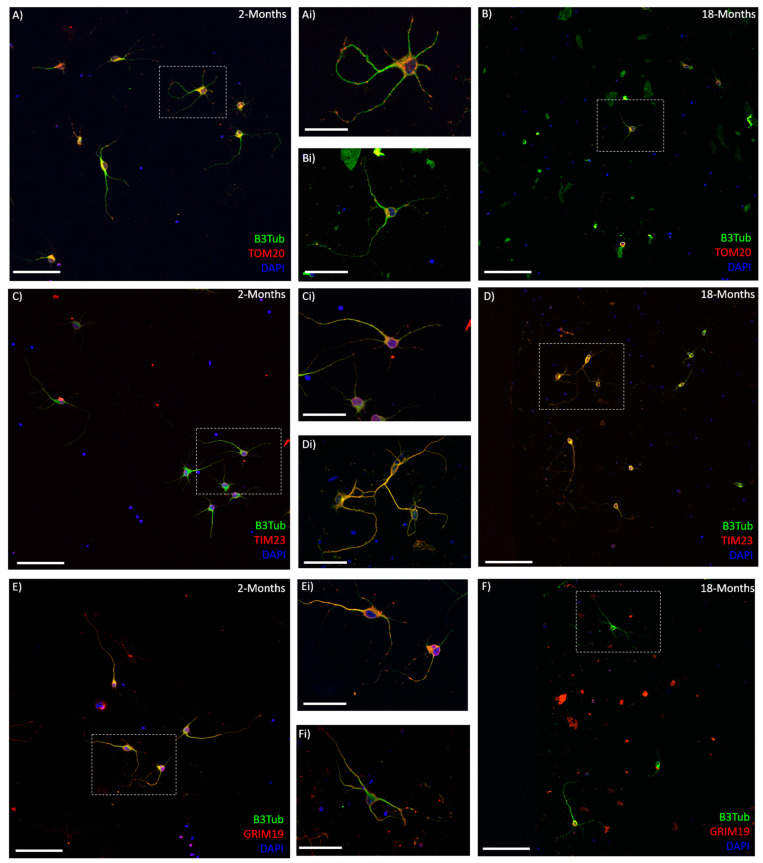
Mitochondrial membrane-associated proteins are altered in cortical neurons with age: (**A**–**F**) representative 20× confocal images of 2-month (**A**,**C**,**E**) and 18-month (**B**,**D**,**F**) cortical neurons after 7 days in vitro, stained with βIII-Tubulin (Green), DAPI (Blue), and either TOM20, TIM23, or GRIM19 (Red). Insets (**Ai**–**Fi**) show higher magnification of single neurons or small clusters in young (**Ai**,**Ci**,**Ei**) and older (**Bi**,**Di**,**Fi**) cultures corresponding to the boxes on (**A**–**F**). Scale bars are 100 μm for (**A**–**F**) and 50 μm for (**Ai**–**Fi**).

## Data Availability

The data presented in this study are available on request from the corresponding author.

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
