# Peer review of "Age-Dependent Decline in Neuron Growth Potential and Mitochondria Functions in Cortical Neurons"

_cells, 2021, doi:10.3390/cells10071625_

Round 1

Reviewer 1 Report

Using a novel method for highly enriched cortical neuronal cultures, the authors compared neuronal mitochondrial functions of mice at 2 months (young), 6-months (young adult), 12-months (middle-aged) and 18-months (old). They chose this model to assess axonal injury and neurite growth in vitro as the basis for all of the subsequent analysis performed in this study. A variety of detrimental changes in neuronal mitochondria in the aging brains were noted that impact the ability to promote neurite growth after injury, although most of the outcome measures were not different statistically, due to low yields and highly variable results. While the results suggest an important role for mitochondria functions in the age-dependent decline in axon growth potential, these collective observations need to be replicated with greater rigor and/or expanded in vivo. In fact, much interpretation of results is offered with very few significant findings. In summary, it appears that the major finding is that there is increased ATP and mtDNA content in aged cortical neurons, but they also have a decreased membrane potential with aging. While many conjectures are offered in the discussion, there are numerous caveats and queries that must be addressed to resolve the insignificant findings, primarily based on improper power analyses and low n/group.

Specific queries for results as presented:

The increase in the mtDNA (and trends for mito proteins) with age that was found to show decrease bioenergetics is indeed unexpected, and therefore requires a future study involving sequencing mtDNA across ages. However, more mtDNA in aged mice implies more mitochondria per cell, but there is no OXPHOS activity (Figure 3B) and very low glycolytic activity (Figure 3I), so what is the compensatory mechanisms by which the cells draw energy?

In Figure 3B, it is unclear whether the aged cells are even alive base on the fact that there is no OXPHOs activity (no response to ADP or FCCP). Moreover, the basal respiration calculation (Fig 3C) doe not match with the OCR trace in Fig 3B. In fact, what does “expected response curve” mean in the legend? Is this actual data or theoretical traces that appear to be misleading?  All OCR seems to be non-mitochondrial in older neurons (Figure 3B). Moreover, their glycolysis is 3 times less than young ones (Figure 3I). Seems like cells are dead in aged groups.

How were OCR graphs generated with 2-6 vs 12-18 mo, despite no significant findings? How/were traces combined?

For Figure 4, the Western bot images are not included, despite their description in legend. Moreover, measuring the activities of the various complexes would validate the low OSPHOS in old neurons; examine activities and not just protein levels.

While it is appreciated why it was measured (Figure 5), ATP levels alone are not very telling without ADP levels. Therefore, the reported insignificant changes in ATP content, without grouping for variability, do not provide a solid foundation to support the authors’ contentions.

For the JC1 experiments to measure membrane potential (Figure 6), the authors need to show max membrane potential (in presence of oligo) and min potential (in presence of FCCP) in one group, at least (truly should be done for all). Without such standardization the values are quite arbitrary.

Immunocytochemistry for several mitochondrial membrane transport proteins show changes in their expression with age (shown in Figure 7), but Figure 8 provides no context as to how the histograms for gray scale measurements were obtained (lacks rigor). Moreover, grouping aged groups to reduced variability is understood, but is it justified statistically in this novel model, in vitro?

The authors are asked to provide rationale and justification as to why certain assays were done only in neurons from female mice.

Statistical analyses:

There seems to be underpowered numbers per group (n=2/group), which may contribute to the high variability and lack of significant differences. In other words, from 2 mice there were 3 triplicates per age, and this seems very underpowered. Moreover, in Figure 8 the N= 50-75/age demonstrates the high variability in this study, likely based upon the low numbers as pointed out.

It does not seem scientifically justifiable to group ‘young’ versus ‘old’, due to small cell yield and low baseline OCR the 2- and 6-month neurons, and the 12- and 18-month neuron. This post-hoc attempt could have been avoided if proper power analysis was done to determine the required sample sizes. Moreover, when N=4/group is stated, this is simply done to increase the number of wells per group.

In the rather lengthy discussion with many conjectures offered based on their findings in vitro, they should compare and contrast their findings to the following studies:

Pandya, J.D., Royland, J.E., MacPhail, R.C., Sullivan, P.G., Kodavanti, P.R., (2016). Age- And Brain Region-Specific Differences in Mitochondrial Bioenergetics in Brown Norway Rats. Neurobiol Aging. Jun;42:25-34.

Yonutas, H.M., Pandya, J.D., Sullivan, P.G., (2015). Changes in Mitochondrial Bioenergetics in the Brain versus Spinal Cord become more Apparent with Age. Journal of Bioenergetics and Biomembranes, 47(1-2), 149-154.

Brown, M.R., Geddes, J.W., Sullivan, P.G., (2004). Brain region-specific, age-related, alterations in mitochondrial responses to elevated calcium. Journal of Bioenergetics and Biomembranes, 36, 401-406.

Author Response

REVIEWER 1

Using a novel method for highly enriched cortical neuronal cultures, the authors compared neuronal mitochondrial functions of mice at 2 months (young), 6-months (young adult), 12-months (middle-aged) and 18-months (old). They chose this model to assess axonal injury and neurite growth in vitro as the basis for all of the subsequent analysis performed in this study. A variety of detrimental changes in neuronal mitochondria in the aging brains were noted that impact the ability to promote neurite growth after injury, although most of the outcome measures were not different statistically, due to low yields and highly variable results. While the results suggest an important role for mitochondria functions in the age-dependent decline in axon growth potential, these collective observations need to be replicated with greater rigor and/or expanded in vivo. In fact, much interpretation of results is offered with very few significant findings. In summary, it appears that the major finding is that there is increased ATP and mtDNA content in aged cortical neurons, but they also have a decreased membrane potential with aging. While many conjectures are offered in the discussion, there are numerous caveats and queries that must be addressed to resolve the insignificant findings, primarily based on improper power analyses and low n/group.

We want to thank the reviewer for the comments. We agree that more work is needed, in vitro first, then in vivo to further characterize the neuron-specific changes in mitochondrial activity with age. However, we believe this work is significant as it is the first assessing the intrinsic regenerative potential of cortical neurons with age and the cell specific (neurons) change in mitochondria activity, to our knowledge. Most of previous literature used preparations from the entire tissue, resulting in the observation of the overall changes in mitochondrial activity, rather than cell- type specific changes. The method used here to enrich cortical neuron allows us to achieve this; however, it provides a limited number of live cortical neurons from each cortex, calling for the need for a high number of mice to be used in each experiment to successfully analyze specific molecules and especially dynamic live cell changes (such as in the Seahorse). This reviewer mentioned that most measures are not statistically significant. The main non-significant outcome observed is associated with the Seahorse data. This is discussed below in detail in response to specific questions from the reviewer, and we agree that having more N for that particular experiment would be of interest, and is part of our future experiments, as we cannot process enough animals simultaneously with our protocol to reach the necessary number of cells, and as we do not currently have access to sufficient 12- or 18-month mice. The other non-significant outcome is the membrane potential measured, with a non-significant reduction observed with age and a p-value of 0.09. This particular experiment used 8 animals per group. We believe that rest of the data presented have sufficient power as we observed significant changes (neurite outgrowth, mtDNA, ATP production, TIM23/GRIM19 protein expression level).

Specific queries for results as presented:

The increase in the mtDNA (and trends for mito proteins) with age that was found to show decrease bioenergetics is indeed unexpected, and therefore requires a future study involving sequencing mtDNA across ages. However, more mtDNA in aged mice implies more mitochondria per cell, but there is no OXPHOS activity (Figure 3B) and very low glycolytic activity (Figure 3I), so what is the compensatory mechanisms by which the cells draw energy?

Our original hypothesis was that mtDNA would be reduced with age in cortical neurons, leading to the reduction of OXPHOS proteins levels. The results obtained were surprising (increase in mtDNA, no change in OXPHOS protein, significant increase in TIM23 and decrease in GRIM19), as noted by the reviewer, but this was significant and consistent.

We agree that further characterization of the neuronal mtDNA changes with age is necessary. While more mitoDNA is generally associated with increase of mitochondria in the cells, this may not be necessary true in the context of aging. There are several studies showing increased mtDNA copy number, but not necessarily increased mitochondrial function, in disease and aging.  For example, patients with mitochondrial disease often show ragged red fibers in their muscle, which is indicative of mitochondrial biogenesis to attempt to compensate for the poor overall mitochondrial function. Recent work from our collaborators has previously shown polymerase gamma (POLG) mutator mice have elevated mtDNA copy number, but decreased OXPHOS function in Seahorse, despite having normal OXPHOS protein (Lei, etal., 2021). This is due to the mutations introduced by the defective polymerase causing the formation of altered proteins. There are many studies demonstrating increased mtDNA point mutations and deletions with aging. It is possible that the reduction of autophagy with age artificially increases the mtDNA levels in the cells, while the mtDNA present is less functional or improperly transcribed to produce functional proteins resulting in no enhancement of mitochondria activity (no increases in the OXPHOS proteins levels). Moreover, it may seem logical that more mtDNA would lead to more OXPHOS protein production and more OXPHOS activity. A potential explanation for the lack of OXPHOS protein increase with the level of mtDNA is that the mtDNA transcription is not as efficient in old neurons as it is in younger neurons, and the same amount of mtDNA leads to less OXPHOS proteins. In this configuration, an increase of mtDNA in older neurons (as we observed) might compensate for this reduction in transcription activity and lead to similar levels of OXPHOS proteins (as we observed). We have clarified this in the discussion (as below). This possibility opens a new area of research, that is to determine the changes of mitochondrial transcriptional activity in neurons with age, and how these can alter mitochondrial functions. In fact, we are currently investigating changes in a particular transcription factor in the mitochondria in cortical neurons with age. On the other hand, it remains possible that, even if the OXPHOS protein levels are not altered with age (potentially because of the increase in mtDNA), these OXPHOS proteins are in fact not as efficient in older mice, as we barely observe any OXPHOS activity in 3B. We are investigating options for future experiments to analyze the activity of each complex separately to further understand these observations.

“…… The increase in mtDNA could also indicate a reduction in mitophagy in older neurons that results in increased numbers of dysfunctional or declining mitochondria retained in the cells. While increased mtDNA is generally associated with an increased number of mitochondria in a cell, it is also possible that the reduction of autophagy with age artificially increases the mtDNA levels. However, the mtDNA is dysfunctional and poorly transcribed. Following this contention, an increase of mtDNA in older neurons might compensate for this reduction in transcriptional activity and lead to similar levels of OXPHOS proteins. A previous study in the brain and spinal cord has demonstrated an increase in mtDNA damage with age, associated with multiple dysfunctions, reductions in NADH/FADH mediated respiration and detrimental increases in ROS [47]. However, this study examined whole CNS tissue as opposed to the cortical neuron cell population examined in the current paper. There have been several previous studies that have shown an increase in mtDNAcn but not necessarily a corresponding increase in mitochondrial function, both in disease [48-50] and aging [51]. A recent study in polymerase gamma (POLG) mutator mice showed elevated mtDNA copy number, but decreased OXPHOS function, despite having normal OXPHOS complex levels [49]..…….”

In Figure 3B, it is unclear whether the aged cells are even alive base on the fact that there is no OXPHOs activity (no response to ADP or FCCP). Moreover, the basal respiration calculation (Fig 3C) doe not match with the OCR trace in Fig 3B. In fact, what does “expected response curve” mean in the legend? Is this actual data or theoretical traces that appear to be misleading?  All OCR seems to be non-mitochondrial in older neurons (Figure 3B). Moreover, their glycolysis is 3 times less than young ones (Figure 3I). Seems like cells are dead in aged groups.

We apologize for lack of clarity, Fig. 3A is theoretical curve (this is now clarified in the legend).

We agree with the reviewer, and we had the same concerns the first time we obtained these data in a test at 2DIV, and in the 4DIV data presented here. However, these neurons were checked under the microscope before beginning the Mito Stress Test assay, and all the wells, with neurons from all the 4 age groups, presented a similar and healthy morphology, and were at the approximately the same density in between wells. Unfortunately, due to time-constraints and the thickness of the plate, we could not take a high-quality image of these before performing the assay. Close inspection of the data suggests small changes in response to the drugs administered but we believe that very little was observed in the older neurons due to the very low basal OCR.

In regard to the differences between the basal rate in Fig 3C compared to 3B; the data presented in the time-course OCR curve in figure 3B is the raw OCR values measured by the Seahorse analyzer over the course of the assay, whereas the Basal rate is calculated based on the last baseline OCR measurement taken before the injection of the first drug and the non-mitochondrial respiration rate. This is clarified in the mat/method and text

“…….. uses the raw OCR and ECAR data collected from the experiment at baseline and after drug application in Wave (Agilent software), and calculates basal and maximal respiration, ATP production, proton leak, non-mitochondrial oxygen consumption, spare respiratory capacity and coupling efficiency.”

This part and the part below make me think we should re-do 1 round of seahorse – either again at 4DIV, or at 2DIV to make sure the aged neurons do not die…would we have everything (mice, seahorse reagents) if we decide to do so?). if we can take a picture before to show the cells look a like (even with a phone!) that would be great.

How were OCR graphs generated with 2-6 vs 12-18 mo, despite no significant findings? How/were traces combined?

We thank the reviewer for this question and apologize for the lack of clarity. The OCR graphs were generated using the Agilent Seahorse report generator for the Mito Stress Test. This is an excel worksheet macro that is generated from the data in the Seahorse WAVE program, used to set up and run the experiment. This macro takes the raw OCR and ECAR data collected from the experiment and calculates all of the factors as below (example from the report generator):

The raw data generated from these calculations for each well was then transferred into Prism for ANOVA and t-test analysis. This report generates graphs from the data and these graphs were compared to those generated in prism to ensure that the data analyzed was the same and allow for the addition of p-values. This has been clarified in the methods:

“……. OCR graphs and functional mitochondrial measures were calculated using the Agilent Seahorse report generator macro for the Mito Stress Test. This macro uses the raw OCR and ECAR data collected from the experiment at baseline and after drug application in Wave (Agilent software), and calculates basal and maximal respiration, ATP production, proton leak, non-mitochondrial oxygen consumption, spare respiratory capacity and coupling efficiency.”

For Figure 4, the Western bot images are not included, despite their description in legend. Moreover, measuring the activities of the various complexes would validate the low OSPHOS in old neurons; examine activities and not just protein levels.

We are sorry this reviewer was not able to access the Western blot images. These were provided separately as a supplementary figure. We hope this is now accessible with the revised manuscript.

We agree that measuring the activities of each OXPHOS complex separately would be an important validation of the protein expression. Currently we are exploring options to test activities of each complex separately (enzymatic assays, Seahorse, targeted drugs), however this is technically challenging with the constraints of aged primary cortical neurons (timing, number of cells extracted per mouse, age of animals, number of animals needed etc.). There is a potential method that we have discussed with the Agilent Seahorse expert by which we could do this, in theory, but we are currently unable to achieve this with our cell constraints.

While it is appreciated why it was measured (Figure 5), ATP levels alone are not very telling without ADP levels. Therefore, the reported insignificant changes in ATP content, without grouping for variability, do not provide a solid foundation to support the authors’ contentions.

We thank the reviewer for this comment. We originally planned to measure the ATP/ADP ratio, as it is important to determine if the cell metabolism is more oxidative or glycolytic. Unfortunately, we were not able to measure ADP levels because of technical challenges. Indeed, all the cell extracts were just enough to perform the ATP assay. Future experiments are necessary to better characterize the age-dependent change in neuronal metabolism, including the ATP/ADP ratio.

For the JC1 experiments to measure membrane potential (Figure 6), the authors need to show max membrane potential (in presence of oligo) and min potential (in presence of FCCP) in one group, at least (truly should be done for all). Without such standardization the values are quite arbitrary.

FCCP treated control cells from the same sample (each sample was split into 2- a control and an experimental), with a minimal membrane potential, were used to normalize all JC-1 measurements. We believe this is common practice and sufficiently normalizes the samples so that any changes can be observed, and represent changes in aging cells. We will investigate the use of oligomycin and FCCP for standardization in future experiments.

Immunocytochemistry for several mitochondrial membrane transport proteins show changes in their expression with age (shown in Figure 7), but Figure 8 provides no context as to how the histograms for gray scale measurements were obtained (lacks rigor).

We apologize for omitting this quantification in the method section. Using ICC the MGV/area of the fluorescent channel representing TOM, TIM, or GRIM was measured for each individual cell. The data represented in Fig. 7 is the mean and SEM of all the individual cells combined.  Fig. 8 provides representative images of these individual neurons (both alone and in detail - 60x- and in the context of the culture -20x) to accompany this data. This has been clarified in the methods.

“……The mean greyscale value (MGV) per area of each individual neuron was analyzed using ImageJ (N= 50-80/age). Specifically, individual neurons, including the cell soma and processes, were manually outlined in ImageJ, and the area and MGV of the AF555 fluorescence was measured for each cell. This was used to calculate the MGV/area of each cell as a measure of fluorescent intensity of the antibody highlighted in the AF555 channel (either TOM20, TIM23, or GRIM19)..…...”

Moreover, grouping aged groups to reduced variability is understood, but is it justified statistically in this novel model, in vitro?

We agree this is a limitation of this paper, at least for the seahorse data. This stems from technical constraints of these experiments (timing, number of cells extracted per mouse, age of animals, number of animals needed etc.) and the availability of the large numbers of aging mice needing to be used at 4 specific ages simultaneously. However, we provide individual age groups as well as pooled data to be fully transparent, and so that the readers can also draw their conclusions.

The authors are asked to provide rationale and justification as to why certain assays were done only in neurons from female mice.

All assays in this paper, except for the mtDNA analysis, were performed using only female mice. This was due to availability, and to keep consistency and avoid any artifact from sex differences. Male mice were only used for the mtDNA analysis to boost sample number, and results were compared between male and female before they were able to be grouped into age-specific analysis.

Statistical analyses:

There seems to be underpowered numbers per group (n=2/group), which may contribute to the high variability and lack of significant differences. In other words, from 2 mice there were 3 triplicates per age, and this seems very underpowered. Moreover, in Figure 8 the N= 50-75/age demonstrates the high variability in this study, likely based upon the low numbers as pointed out.

We agree with the reviewer that some experiments (specifically, the RT-qPCR purity and Seahorse OCR analysis) are underpowered, while the majority have sufficient N. While we would like to further increase the N for the Seahorse, this is technically difficult due to time, age and number of mice constraints.  The N=2 used for the purity analysis was due to the amount of RNA able to be extracted from the cells after Seahorse analysis, and in fact represents large numbers of mice. The Seahorse analysis used combined 2- & 6-month and 12- & 18-month groups due to technical and mouse constraints, to increase the N close to statistical power. While successful, adult cortical neuron purification leads to a very limited number of cells, which decreases with age necessitating the use of a high number of older mice.

The data represented in Fig 7 & 8 is analyzed on an individual cell basis and, as such, represents an equivalent of an N of 50-80 for each of the 4 age groups. We contend that this is a very substantial N, especially when these are then grouped further into young and old (creating an N of 100+). Within this N we can see highly significant differences (P<0.0001). This suggests that the differences observed in these immunocytochemistry measurements were very substantial and where there is no statistical significance, there is a valid trend but insubstantial difference.

It does not seem scientifically justifiable to group ‘young’ versus ‘old’, due to small cell yield and low baseline OCR the 2- and 6-month neurons, and the 12- and 18-month neuron. This post-hoc attempt could have been avoided if proper power analysis was done to determine the required sample sizes. Moreover, when N=4/group is stated, this is simply done to increase the number of wells per group.

Here again we agree with the reviewer. This is a limitation. With the cell extraction method, the best developed so far for purifying/culturing cortical neurons, the number of cells obtained per mouse is low, and even lower when using older mice. With a proper power analysis, we estimate the need of 5 wells (at a minimum of 50,000 cells/well) per group, which would mean approximately 10 mice in each age group (or in young/old). This was both technically difficult to achieve, as with this system we can process a maximum of 8 mice simultaneously at the moment (2 mice for each group to be plated at the same time on the same plate), and an inaccessible number of aging mice. While we’re developing the technique to process higher number of mice at once and investigating other approaches to assess cortical neuron mitochondrial changes with age, we are not able to reach that number of needed animals.

In the rather lengthy discussion with many conjectures offered based on their findings in vitro, they should compare and contrast their findings to the following studies:

The discussion has been edited and reduced. These suggested papers have been included in the discussion:

Pandya, J.D., Royland, J.E., MacPhail, R.C., Sullivan, P.G., Kodavanti, P.R., (2016). Age- And Brain Region-Specific Differences in Mitochondrial Bioenergetics in Brown Norway Rats. Neurobiol Aging. Jun;42:25-34.

This paper is cited in the discussion. We would be interested to attempt an enzymatic analysis such as the one preformed in Pandya et al. on our isolated neuron population. This paper used mitochondrial isolated from whole tissue in specific regions of the brain. We have yet to attempt to isolate mitochondria from our specific cell population as the starting samples are likely too small in their current state. This would be interesting to attempt, given sufficient mice to isolate sufficient neurons for mitochondrial extraction. This paper used mitochondria extracted from entire brain regions, and thus all the cell populations present in that region, as opposed to our use of a single isolated cell type.

“……… . A previous study using the Seahorse system to measure OCR from mitochondria isolated from different brain regions also demonstrated a reduction in OCR with age in rats. However, the alterations to respiratory complexes with age demonstrated in this previous paper were region-specific and combined the mitochondria from all cell populations present [63]. …….. Further investigation of each individual OXPHOS complex and their enzymatic activity in cortical neurons (as previously analyzed in isolated mitochondria from whole brain tissue [63]) would be helpful to answer the questions raised by these results.…….”

Yonutas, H.M., Pandya, J.D., Sullivan, P.G., (2015). Changes in Mitochondrial Bioenergetics in the Brain versus Spinal Cord become more Apparent with Age. Journal of Bioenergetics and Biomembranes, 47(1-2), 149-154.

This paper was previously cited in relation to increased mtDNA damage with age, but not explored in detail due to the length of the discussion. It shows increased mtDNA in the aging CNS, as well as respiratory dysfunction and increased ROS. This could be linked to our conclusions so far, as aging cortical neurons had increased mtDNA but dysfunctional respiration. This is addressed in the discussion of this paper here Yonutas et al. is cited.

“…….. The increase in mtDNA could also indicate a reduction in mitophagy in older neurons that results in increased numbers of dysfunctional or declining mitochondria retained in the cells. While increased mtDNA is generally associated with an increased number of mitochondria in a cell, it is also possible that the reduction of autophagy with age artificially increases the mtDNA levels. However, the mtDNA is dysfunctional and poorly transcribed. Following this contention, an increase of mtDNA in older neurons might compensate for this reduction in transcriptional activity and lead to similar levels of OXPHOS proteins. A previous study in the brain and spinal cord has demonstrated an increase in mtDNA damage with age, associated with multiple dysfunctions, reductions in NADH/FADH mediated respiration and detrimental increases in ROS [47]. However, this study examined whole CNS tissue as opposed to the cortical neuron cell population examined in the current paper. …..….”      

Brown, M.R., Geddes, J.W., Sullivan, P.G., (2004). Brain region-specific, age-related, alterations in mitochondrial responses to elevated calcium. Journal of Bioenergetics and Biomembranes, 36, 401-406.

We acknowledge that calcium balance is an important facet of mitochondria, however, in the current manuscript we do not specifically examine calcium at all. This suggested paper is mentioned in both the introduction and discussion.

            “….. It has previously been observed that Ca2+ buffering is impaired in the aging brain [15]…”

“……Previously, mitochondria extracted from whole brain tissue have shown a region-specific increase in ROS production as well as impaired Ca2+ buffering [15]. …..”

Reviewer 2 Report

This study examined the neurite outgrowth potential of cultured, cortical neurones obtained from mice ages 2, 6, 12 and 18 months. Cultured neurones were maintained in vitro for several days and the extent of growth imaged using antibody staining. Moreover, the authors report that older neurones exhibited dysfunctional respiratory potential with changes to mitochondrial member potentials and transport proteins. Whereas, on the other hand, the mitochondrial DNA abundance and cellular ATP levels were increased. This study concluded that it is mitochondrial dysfunction in the older neurones (age-related) that may corroborate with poor growth potential after spinal cord injury.

Major Issues:

The issue of seeking and identifying factors that may promote nerve regeneration is important and the research interesting. This study, however has significant issues both in the context of writing- style, logical of inquiry and rational; these would need to be significantly revised prior to any serious review of this paper. This manuscript is poorly written, with fragmented sentence  structure - starting with the abstract (e.g. How it is impacted by aging impact this is unknown ---. Using isolation and culture of adult ---) and throughout. The rationale behind this research is also not correct as this paper has nothing to do with spinal cord nerve injury or traumatic brain injury. For this to have occurred they should have used appropriate models of injury and not rely on the poor growth/regeneration potential of adult neurones. It is important to note that in humans, the adult CNS neurones do not regenerate; for this the authors should have used spinal cord (PNS) neurones which do exhibit the growth potential. Thus the translation potential of this research is also oversold.  It would have helped if the authors could identify the precise cortical regions from where the neurones were isolated. For regeneration after spinal injury, they should have instead used DRG neurones and then ask the questions that they had attempted to answer while using cortical neurones.

Moreover, they claim that this is the first study to elude towards the possibility of mitochondrial involvement in nerve injury and regeneration; to which I highlight some important omissions in this study.

Wang et al., 2021 Front. Aging Neurosi,

Smith and Gallo, 2018, Develop. Neurobiol.

Zhou et al., (2016 JCB). Facilitation of axon regeneration by enhancing mitochondrial transport and rescuing energy deficits.

“Although neuronal regeneration is a highly energy-demanding process, axonal mitochondrial transport progressively declines with maturation. Mature neuronse typically fail to regenerate after injury, thus raising a fundamental question as to whether mitochondrial transport is necessary to meet enhanced metabolic requirements during regeneration. Here, we reveal that reduced mitochondrial motility and energy deficits in injured axons are intrinsic mechanisms controlling regrowth in mature neurones. Axotomy induces acute mitochondrial depolarization and ATP depletion in injured axons. Thus, mature neurone-associated increases in mitochondria-anchoring protein syntaphilin (SNPH) and decreases in mitochondrial transport cause local energy deficits. Strikingly, enhancing mitochondrial transport via genetic manipulation facilitates regenerative capacity by replenishing healthy mitochondria in injured axons, thereby rescuing energy deficits. An in vivo sciatic nerve crush study further shows that enhanced mitochondrial transport in snph knockout mice accelerates axon regeneration. Understanding deficits in mitochondrial trafficking and energy supply in injured axons of mature neurones benefits development of new strategies to stimulate axon regeneration.”

Han et al; 2016 Neuron (Mitochondria localize to injured axons to support regeneration)

“Axon regeneration is essential to restore the nervous system after axon injury. However, the neuronal cell biology that underlies axon regeneration is incompletely understood. Here we use in vivo, single-neurone analysis to investigate the relationship between nerve injury, mitochondrial localization, and axon regeneration. Mitochondria translocate into injured axons so that average mitochondria density increases after injury. Moreover, single-neurone analysis reveals that axons that fail to increase mitochondria have poor regeneration. Experimental alterations to axonal mitochondrial distribution or mitochondrial respiratory chain function result in corresponding changes to regeneration outcomes. Axonal mitochondria are specifically required for growth-cone migration, identifying a key energy challenge for injured neurones. Finally, mitochondrial localization to the axon after injury is regulated in part by dual-leucine zipper kinase 1 (DLK-1), a conserved regulator of axon regeneration. These data identify regulation of axonal mitochondria as a new cell-biological mechanism that helps determine the regenerative response of injured neurones”.

This study thus did not invoke an appropriate model to test for the role of mitochondria in nerve regeneration nor was it a better fit to test their hypothesis. It is important to note that there is a tremendous difference between nerve injury and studying dissociated neurons in culture. In vitro cultured neurons are subjected to complete denervation (recapitulation of neurodevelopment program and switching on of all genes), enzymatic treatment, physical dissociation – all of these are not reflective of nerve injury. Moreover, the neurons are also maintained in an unnatural environment, as such parallels cannot be drawn. It had better if the authors stuck to culture neurons and examined their growth potential on various days. Also they must have included appropriate controls using neurons from P0 animals. They should have also explicitly stated the culture conditions (number of dishes, numbers of neurons/dish, % of cells sprouted, cellular viability, dead-live assay etc.), and whether these experiments were done blind or not. In short, their model would be more befitting with the narrative of “neuronal aging in culture and the role of mitochondria” with an added variable of animals from various ages. It is also not clear why only female animals were used as their hormonal cycles would vary with ages and thus could have been another confounder.

Introduction and discussion sections need considerable edits as well – there are run on sentences and fragmented structures. The discussion is too long and mostly irrelevant and should be condensed to at least 60%.

In the result section, it is also not clear if decreased mitochondrial function was due to reduced branching or if the branching was reduced due to mitochondrial dysfunction (cause or causal).

A fair comparison would have been to include control neurons from spinal cord and P0 animals; without these data, I am not sure what could be the overall significance and importance of this work. Moreover, the mitochondrial changes only show a trend but there was no significant difference which would mean that either the authors increase their n’ values or to discussion why they did not see any significant changes.

In summary, there are some interesting data in this paper that could be importance to the readership of this journal but it would need to be packaged more appropriately and realistically.

Author Response

This study examined the neurite outgrowth potential of cultured, cortical neurones obtained from mice ages 2, 6, 12 and 18 months. Cultured neurones were maintained in vitro for several days and the extent of growth imaged using antibody staining. Moreover, the authors report that older neurones exhibited dysfunctional respiratory potential with changes to mitochondrial member potentials and transport proteins. Whereas, on the other hand, the mitochondrial DNA abundance and cellular ATP levels were increased. This study concluded that it is mitochondrial dysfunction in the older neurones (age-related) that may corroborate with poor growth potential after spinal cord injury. 

Major Issues:

The issue of seeking and identifying factors that may promote nerve regeneration is important and the research interesting. This study, however has significant issues both in the context of writing- style, logical of inquiry and rational; these would need to be significantly revised prior to any serious review of this paper. This manuscript is poorly written, with fragmented sentence  structure - starting with the abstract (e.g. How it is impacted by aging impact this is unknown ---. Using isolation and culture of adult ---) and throughout.

We regret that this reviewer finds this manuscript poorly written. We agree that we missed some typos and poorly structured sentences during the editing. We have modified the text accordingly

The rationale behind this research is also not correct as this paper has nothing to do with spinal cord nerve injury or traumatic brain injury.

We have to completely disagree with the reviewer on this point for several reasons. The corticospinal tract (CST) projecting from the motor cortex to the spinal cord is one the most important tracts controlling the movements of the upper and lower limbs (Welniarz, Dusart & Roze, 2017). This tract is not regenerating after brain or spinal cord injury, which is a prominent cause of paralysis. In fact, it is the most refractory to regeneration, while other tracts (rubrospinal tract etc..) have some regenerative potential. Understanding why the CST is refractory to regeneration, and finding ways to promote its regeneration, is the grail in the spinal cord injury field. Therefore, understanding how mitochondrial activity, which we know are involved in axon growth in non-cortical neurons as indicated by the reviewer, is of high importance to develop strategies to promote axon regeneration.

In this report, we are concentrating on central nervous neurons, not peripheral neurons (such as the spinal cord nerve mentioned by the reviewer), that are damaged by SCI and are integral to functional recovery after an injury to the spinal cord.

For this to have occurred they should have used appropriate models of injury and not rely on the poor growth/regeneration potential of adult neurones.

Assessing the impact of injury on the cortical neurons mitochondrial function in vivo is of high interest. We have some plans for ongoing research on the topic and the data presented here are the first in vitro step to demonstrate this is an avenue of research worth pursuing. Moreover, the focus of this paper is on the impact that aging has on neuronal growth and regeneration, rather than on development, and the observed decline in neuron growth potential with age, irrespective of injury.  The increasing age of the SCI population is a great concern that has not been strongly addressed in the field and understanding the impacts of aging alone is an important step towards remedying this. 

It is important to note that in humans, the adult CNS neurones do not regenerate; for this the authors should have used spinal cord (PNS) neurones which do exhibit the growth potential.

This is absolutely correct, the adult CNS (in mammals in general) does not regenerate in vivo. This regenerative potential is even further reduced with age. This is exactly why studying the changes in the regenerative potential with age, in cortical neurons, is of high interest. We totally agree that assessing the regenerative potential with age spinal neurons is of high interest. However, to date, there are very limited methods to culture embryonic, postnatal, and young adult spinal neurons (Beudet, et al., 2015; Eldeiry, et al., 2017), and no success in aged primary neurons. While we are attempting this with our new methodology, and while we have some success in culturing young adult spinal neurons, the number of neurons is extremely low and is not compatible with most of the techniques used in this manuscript to assess the impact of age on mitochondrial activity.

Thus the translation potential of this research is also oversold. 

While did not propose any direct translation potential of this work to clinical testing, our intention is the suggest that understanding neuronal mitochondrial dysfunctions occurring with age is the first step toward exploring the promotion of this activity in the interest of promoting regenerative potential in older neurons. Several sentences have been modified to reflect this intent more obviously, and hopeful allay any confusion.

“……. Collectively, our results suggest an important role for mitochondrial functions in the age-dependent decline in axon growth potential. This also may suggest the exploration of promoting neuronal mitochondrial function to promote axon growth in injured neurons regardless of age or time post injury as a promising future direction. These observations need to be expanded in vivo and strategies to manipulate the mitochondria need to be analyzed in detail.”  

“…….. This bears greater scrutiny in order to target mitochondria in aging neurons with the eventual goal of pursuing a potentially efficacious therapy for SCI in patients of all ages.”

It would have helped if the authors could identify the precise cortical regions from where the neurones were isolated.

We thank the reviewer for this comment, and we completely agree. However, there are significant technical challenges to doing so, including the fact that we would not obtain sufficient numbers of neurons to perform any of the experiments we did. These cultures and preparations contain all the cortical neurons. Even if we could separate different regions, it would be difficult to distinguish the different cortical layers. Another approach would be to look directly in the cortex using immunofluorescence. While these experiments are ongoing, they are beyond the goal of this manuscript.

For regeneration after spinal injury, they should have instead used DRG neurones and then ask the questions that they had attempted to answer while using cortical neurones.

DRG neurons, part of the peripheral nervous system, are known to regenerate in the nerve, and eventually in the spinal cord, but only after a conditioning lesion of the nerve (Neumann & Woolf, 1999). Additionally, regeneration of the DRG axons will not impact locomotor recovery after spinal injury, as it is control by supraspinal neurons. Finally, culture of DRG neurons, from any age, has been possible for a long time and assessing the impact of age on DRG neurons mitochondrial activity would not be as novel, or informative to our purposes, as assessing the impact of age on cortical neurons mitochondrial activity

Moreover, they claim that this is the first study to elude towards the possibility of mitochondrial involvement in nerve injury and regeneration; to which I highlight some important omissions in this study.

We thank the reviewer for pointing this out. It seems our message was not clear. First, we are focusing on cortical neurons, CNS neurons, not nerve or peripheral nervous system neurons. Second, we do not make the claim that this is the first study to elude towards the possibility of mitochondrial involvement in regeneration. We link the decline in neuron growth potential with dysfunctional mitochondria with a specific focus on aging. The novel element of this study is linked to the assessment of mitochondria specifically and only in aging cortical neurons, and the relationship between that and decreased neurite growth. Here are some examples of the claims in this study:

  • “Collectively, our results suggest an important role for mitochondrial functions in the age-dependent decline in axon growth potential.”
  • “This study represents the first comparison of primary cortical neurons isolated from mice at different ages.”
  • “This work is also the first to specifically address changes in mitochondrial functions with age in cortical neurons.”
  • “This clearly demonstrates the age-dependent decline in cortical neurons neurite outgrowth in vitro, suggesting that neuron-intrinsic properties are involved in this reduction.”
  • “The mitochondrial alterations we have observed in older neurons may play a significant role in the age-dependent decline in axon growth potential.”

We thank the reviewer for providing the following references. All of these were already mentioned in the manuscript.

Wang et al., 2021 Front. Aging Neurosi,

Smith and Gallo, 2018, Develop. Neurobiol.

Both papers were previously read and considered, and are referenced in both the introduction and discussion (however, not in detail):

  • “Mitochondria play important roles in normal aging [9], axon growth [25-30], and the progression of SCI [31-33].”
  • “Mitochondria are vital to axon growth and regeneration [29, 30]. ………”

Zhou et al., (2016 JCB). Facilitation of axon regeneration by enhancing mitochondrial transport and rescuing energy deficits.

This paper was already referenced and highlighted specifically in the discussion:

  • “……… A recent study using a syntaphilin knock-out mouse model, lacking the static anchor protein to hold axonal mitochondria stationary and resulting in increased axonal mitochondrial motility, showed improvements in three distinct models of SCI and axonal injury [87]. This model suggested that enhancing mitochondrial transport and motility can enhance recovery after SCI in young mice [87]. However, this contention has yet to be tested in an aging paradigm. The regulatory mechanisms that transport, distribute, and clear mitochondria in neurons are compromised in neurotrauma [88], which may compound the retention of dysfunctional mitochondria that is seen in aging. Mitochondrial energetics are beneficial to axon regeneration in SCI in young mice [87], however this has not been examined in middle-aged and aging animals. ………”

“Although neuronal regeneration is a highly energy-demanding process, axonal mitochondrial transport progressively declines with maturation. Mature neuronse typically fail to regenerate after injury, thus raising a fundamental question as to whether mitochondrial transport is necessary to meet enhanced metabolic requirements during regeneration. Here, we reveal that reduced mitochondrial motility and energy deficits in injured axons are intrinsic mechanisms controlling regrowth in mature neurones. Axotomy induces acute mitochondrial depolarization and ATP depletion in injured axons. Thus, mature neurone-associated increases in mitochondria-anchoring protein syntaphilin (SNPH) and decreases in mitochondrial transport cause local energy deficits. Strikingly, enhancing mitochondrial transport via genetic manipulation facilitates regenerative capacity by replenishing healthy mitochondria in injured axons, thereby rescuing energy deficits. An in vivo sciatic nerve crush study further shows that enhanced mitochondrial transport in snph knockout mice accelerates axon regeneration. Understanding deficits in mitochondrial trafficking and energy supply in injured axons of mature neurones benefits development of new strategies to stimulate axon regeneration.”

 Han et al; 2016 Neuron (Mitochondria localize to injured axons to support regeneration)

This paper was already referenced in the introduction and discussion:

  • “…… Mitochondria play important roles in normal aging [9], axon growth [25-30], and the progression of SCI [31-33]. ……..”
  • “……. After CNS injury the mitochondria translocate into the injured axons resulting in an increase in the mitochondria density [25]. A failure to increase mitochondrial function in neurons is linked to poor regeneration [25]. …….”

“Axon regeneration is essential to restore the nervous system after axon injury. However, the neuronal cell biology that underlies axon regeneration is incompletely understood. Here we use in vivo, single-neurone analysis to investigate the relationship between nerve injury, mitochondrial localization, and axon regeneration. Mitochondria translocate into injured axons so that average mitochondria density increases after injury. Moreover, single-neurone analysis reveals that axons that fail to increase mitochondria have poor regeneration. Experimental alterations to axonal mitochondrial distribution or mitochondrial respiratory chain function result in corresponding changes to regeneration outcomes. Axonal mitochondria are specifically required for growth-cone migration, identifying a key energy challenge for injured neurones. Finally, mitochondrial localization to the axon after injury is regulated in part by dual-leucine zipper kinase 1 (DLK-1), a conserved regulator of axon regeneration. These data identify regulation of axonal mitochondria as a new cell-biological mechanism that helps determine the regenerative response of injured neurones”.

This study thus did not invoke an appropriate model to test for the role of mitochondria in nerve regeneration nor was it a better fit to test their hypothesis.

It is true that we did not assess change in cortical neurons mitochondrial functions occurring after brain or spinal cord injury. We only focused on the changes occurring with age. Change in mitochondrial functions in the context of CNS injury (not nerve injury) is the focus of ongoing project in the laboratory and a logical extension from the data presented in this manuscript, as suggested by this reviewer.

It is important to note that there is a tremendous difference between nerve injury and studying dissociated neurons in culture. In vitro cultured neurons are subjected to complete denervation (recapitulation of neurodevelopment program and switching on of all genes), enzymatic treatment, physical dissociation – all of these are not reflective of nerve injury. Moreover, the neurons are also maintained in an unnatural environment, as such parallels cannot be drawn. It had better if the authors stuck to culture neurons and examined their growth potential on various days.

The reviewer is absolutely right, in vitro and in vivo setting are different. As of now, it is technically impossible to directly assess mitochondrial respiration, membrane potential or ATP production of specific cells in vivo. There have been some strides made in assessing these in situ using brain slices and slice cultures, however this is not cell-type specific. Dissociation of neurons, while traumatic, does not mimic a real brain or spinal injury. Similarly, in vitro culture can artificially alter the neurons. In several experiments (fig. 2, 4-6) un-cultured neurons were used directly after dissociation in an attempt to stay as close as possible to the in vivo settings. While these neurons may respond with the short period necessary for dissociation and measurement, we believe the difference in responses with age is a demonstration of the impact of age on mitochondrial activity. We wanted to unequivocally assess the changes in cortical neurons. One could certainly advance the reverse when assessing changes in vivo directly. In the complex environment of the spinal cord, it may be that extrinsic factors, rather than the neuron-intrinsic capacities, are altering the mitochondrial activity.

Also they must have included appropriate controls using neurons from P0 animals. They should have also explicitly stated the culture conditions (number of dishes, numbers of neurons/dish, % of cells sprouted, cellular viability, dead-live assay etc.), and whether these experiments were done blind or not. In short, their model would be more befitting with the narrative of “neuronal aging in culture and the role of mitochondria” with an added variable of animals from various ages. It is also not clear why only female animals were used as their hormonal cycles would vary with ages and thus could have been another confounder.

We’re sorry that the reviewer feels we need to include more controls, “variable of animals from various ages”. We are already comparing 4 different adult age groups, from 2-months to 18-month old animals to demonstrate changes in mitochondrial activity in adult cortical neurons. Using P0 neurons, which are actively growing axons and will subsequently have strong mitochondrial activity (Steketee, et al., 2012; Flippo and Strack, 2017) will not alter our conclusions in any ways and will not address the main question: changes in mitochondrial activity with age in adult cortical neurons.

All the quantifications were done blindly by students and collaborators. This is now mentioned in the methods.

Females were largely used for this study due to availability in the lab and to avoid any artifact from sex differences. To minimize mitochondrial changes that could be associated with hormonal cycle, mice were visually inspected and only used when not in their estrus cycle (looking at the color, swelling and size of the vaginal opening). It is true, as mentioned by the reviewer, that mice in our 12 and 18months group may have become acyclic (pre-menopause). However, any systematic hormonal influence stemming from age is not likely to have significant lasting effect once the cells are in a controlled in vitro situation for several days. However, we acknowledge that hormonal changes with age may be a contributing factor to age-dependent alterations in mitochondrial activity. There have been previous observations linking changes in estrogen to mitochondrial changes in various tissues (Torres, et al., 2018), including the brain (Torrens-Mas, et al., 2020). In the brain, estrogen plays neuroprotective, neurotrophic and antioxidant roles. Changes in estrogen in menopause have been associated with mitochondrial alterations and may increase vulnerability to brain degeneration and age-related pathologies (Lejri, et al., 2018). Mitochondrial dysfunctions in various diseases have also been linked to alterations in estrogen (Mooga, et al., 2018). Taking this into consideration, potential hormonal changes with age might in fact be a contributing factor to the changes in mitochondrial activity in neurons, rather than a confounding factor. This hypothesis would have to be specifically tested in the future. 

Introduction and discussion sections need considerable edits as well – there are run on sentences and fragmented structures. The discussion is too long and mostly irrelevant and should be condensed to at least 60%.

We agree that the discussion was long.  We have now performed a more detailed editing, by authors and by a professional editor and significantly reduced the discussion.

In the result section, it is also not clear if decreased mitochondrial function was due to reduced branching or if the branching was reduced due to mitochondrial dysfunction (cause or causal).

We have not demonstrated any causality – it would be interesting to promote mitochondrial activity, genetically or pharmacologically, to answer this question. We are currently working on manipulating mitochondria in these cells as a next step from the observations of this study. We thank the reviewer for pointing this out.

A fair comparison would have been to include control neurons from spinal cord and P0 animals; without these data, I am not sure what could be the overall significance and importance of this work. Moreover, the mitochondrial changes only show a trend but there was no significant difference which would mean that either the authors increase their n’ values or to discussion why they did not see any significant changes.

As mentioned above, the goal of this work is to assess changes in mitochondrial activity in adult cortical neurons with age. Comparing to postnatal animals, with actively growing and developing CNS, would not provide any further interest or progression towards answering the central questions of this paper. We see significant changes in many experiments throughout the manuscript. Moreover, slight changes in mitochondrial activity, even if only a trend and not significant, can have tremendous consequences in the overall health of the neurons. Because of the difficult nature and timing of the performed experiments (age of the animals, isolation of viable live cells), and the number of mice required to produce samples able to be analyzed by these methods, adding N values is extremely challenging.

In summary, there are some interesting data in this paper that could be importance to the readership of this journal but it would need to be packaged more appropriately and realistically.

Reviewer 3 Report

The manuscript described a study revealing the association between the age-dependent decline in neuron growth potential in the CNS and the age-related dysfunction of neuronal mitochondria by comparing four ages of mice- 2- months (young), 6-months (young adult), 12-months (middle-aged) and 18-months (old). This study contains some observations that may be interesting to general readers.

  1. It’s a little confusing. Is the mouse used in this study a SCI mouse model? Since the authors mentioned that this study is related to SCI.
  2. How the total neurite length was exactly measured and calculated? Does the average neurite length equal the total neurite length/cell number?
  3. It would be interesting to see the comparison of mitochondrial ROS levels as well as the mitochondrial DNA integrity (by LA-PCR) of all four groups, two critical factors related to mitochondria functions, as the authors mentioned in the text.
  4. Strongly suggest narrowing down the Discussion to like half to make it more focus.

Author Response

The manuscript described a study revealing the association between the age-dependent decline in neuron growth potential in the CNS and the age-related dysfunction of neuronal mitochondria by comparing four ages of mice- 2- months (young), 6-months (young adult), 12-months (middle-aged) and 18-months (old). This study contains some observations that may be interesting to general readers.

  1. It’s a little confusing. Is the mouse used in this study a SCI mouse model? Since the authors mentioned that this study is related to SCI.

The mice used in this study were wild-type with no injury.

In SCI the cortico-spinal tract (CST), descending from the motor cortex, is disrupted. Disruption of this tract is largely responsible for the resulting paralysis or movement deficits. This tract is not regenerating SCI. In fact, it is the most refractory to regeneration, while other tracts (rubrospinal tract etc..) have some regenerative potential. In this paper, we look at the effects of aging on mitochondria in the cortical neurons that project axons into the CST; that is, from the cortex. We use an isolated enriched population of cortical neurons to specifically examine the changes in mitochondria over different ages. We are specifically focusing on the aging factor in this paper as the SCI population demographic is changing, getting older, and this has not received sufficient attention in research to date. The next step is to include SCI into the parameters, however, for this initial exploration, we did not want to confound the aging changes with the injury changes.

  1. How the total neurite length was exactly measured and calculated? Does the average neurite length equal the total neurite length/cell number?

The neurites were measured by hand using the tracing macro for ImageJ, NeuronJ. This gave measurements in μm based on the μm/pixel scaling of the images. This macro measures the length of each individual neurite traced. The tracings are grouped per cell so the sum of all the neurites and the average length taken from all the neurites on a particular cell are also calculated. All of this analysis is performed cell by cell, so the average neurite length represents the average length of a neurite on an individual cell.

  1. It would be interesting to see the comparison of mitochondrial ROS levels as well as the mitochondrial DNA integrity (by LA-PCR) of all four groups, two critical factors related to mitochondria functions, as the authors mentioned in the text.

We agree with the reviewer that this is potentially very important information. We are currently optimizing the use of CellROX to analyze the ROS levels in these particular cells in vitro, and exploring other avenues to test mitochondrial ROS production in our primary cortical neurons. While preliminary data suggests change in ROS production in older cortical neurons, these are still pilot data and need to be confirmed before making definitive conclusions. Additionally, the use of LA-PCR  (Gonzalez-Hunt, et al., 2016) will be a great alternative to further understand the mitochondrial DNA changes occurring with age in cortical neurons. Unfortunately, these will be followed up experiments, as we currently aging mice in colony to perform these suggestions.

  1. Strongly suggest narrowing down the Discussion to like half to make it more focus.

The discussion has been edited and reduced for greater clarity.

Reviewer 4 Report

This is an interesting report about the age dependent decline in the mitochondrial function. 

1. The major weak point of the study is the largely differing neuronal enrichment between days 2 and 4 in culture:

The neuronal enrichment between 2 days in culture and 4 days in culture differed by the factor of 8 (CT difference between days 2 and 4 is about 3 as stated in § 3.1). 

The authors should therefore provide for all presented experiments the enrichment measurements (according to 3.1) to convince the reader, that the results are not obscured by differences in glial cell contamination.

That could perhaps explain the very large differences in the seahorse measurements presented in Fig. 3D and the rather small differences in the protein expression data (Fig. 4).

2. For the presented bar graphs in Fig. 3 the results of extensive statistical testing should be given. It is not clear, if some of the big differences (cf. Fig. 3D, mean difference more than five fold) are significant or not. 

3. The observed higher ATP levels in older neurons are contradicting the lower respiration rates (poorer RC performance) and poorer coupling (lower membrane potential) of mitochondria. This could be a result of composition changes during culture (cf. point 1).

4. The English requires a very careful editing.

Author Response

Comments and Suggestions for Authors

This is an interesting report about the age dependent decline in the mitochondrial function. 

 We thank the reviewer for finding this report interesting.

  1. The major weak point of the study is the largely differing neuronal enrichment between days 2 and 4 in culture:

The neuronal enrichment between 2 days in culture and 4 days in culture differed by the factor of 8 (CT difference between days 2 and 4 is about 3 as stated in § 3.1). 

We thank the reviewer for pointing this out. It is true that the final CT values calculated were 10.123 at D0 (right after cell preparation), 13.074 at D2 and 11.051 at D4. While we observed a CT value difference of about 2 between D2 and D4, we want to emphasize that the CT value of 10 or 13 is synonymous of highly enriched neuronal population compared to glial population. In fact, the RT-qPCR results for young and old samples were very similar between ages. That said, the decrease in CT value between D2 and D4, and therefore the potential decrease in neuronal enrichment of the samples, is likely to be the result of neuronal cell loss in culture and due to the Seahorse analysis performed. These cortical neurons are very delicate in culture, while any stray astrocytes that remained in the sample are much hardier cells and have much greater chance of survival, especially after being subjected to multiple drug manipulations and analysis. Similarly, the difference between D0 and D2 could be explained by the fact that right after cell dissociation, there are slightly more glial cells than at D2, and the culture conditions reduce the pool of glial cells, facilitating neuron survival, further enriching the cell culture toward the neuron population. However, the dissociation protocol enriches the neuronal population by an impressive CT value of 10, consistently.  This clearly demonstrates that these preparations have a minimal number of glial cells and make us confident that all the samples analyzed when collected directly after isolation with no culture (cell lysate, RNA, Protein, DNA) represent the cortical neuron population. Similarly, any confounding changes or cell loss in culture will not have any effect on the data, as the CT value is even higher than D0, suggesting an even more pure neuronal enrichment.

The authors should therefore provide for all presented experiments the enrichment measurements (according to 3.1) to convince the reader, that the results are not obscured by differences in glial cell contamination.

We agree that systematic qRT-PCR would be fantastic, but we did not perform this as it is technically challenging. Indeed, the entire cell preparation is required in most of the experiments for protein extraction, cell lysate etc., while the entire cell preparation is also required for extracting enough RNA for the qRT-PCR. Additionally, our immunocytochemistry data on cultured cells shows that the majority of cells (~80%) express βIII-Tubulin, while only a very minor population expressing GFAP (see below) and no oligodendrocytes were observed (Olig2) in our culture conditions. This supports our surmise that any possible glial contamination is extremely low. Additionally, all the data presented here using immunocytochemistry is specific to βIII-Tubulin expressing neurons (Fig 1 and 8). This has been added into the text of this paper:

“………RNA extracted from the neuron enriched negative fraction immediately following MACS dissociation and isolation (D0) showed a neuron enrichment -ΔCT of 10.123. After 4-days in vitro with neuron specific culture methods, and analysis using the Agilent Seahorse, the neuron-enriched population had a -ΔCT of 11.051. This corroborates preliminary culture data (unpublished) where we observed 81.5± 8.2% βIII-Tubulin positive neurons and <2±1.5% cells expressing GFAP. Olig2 expression was not observed in samples from 2-days and 4-days in vitro. Following from this we are confident the data presented below are from a highly enriched neuronal population.. ……..”

That could perhaps explain the very large differences in the seahorse measurements presented in Fig. 3D and the rather small differences in the protein expression data (Fig. 4).

As discussed above, the CT value at d0, d2 or d4 demonstrate a high increase in the neuronal population compared to glial cells, despite some variations between the days of the preparation. Additionally, while the protein expression showed no significant alterations with age, the trending differences observed in the Seahorse measurements were small and also not significant. This leads us to suspect that any potential glial contamination is not a significant factor in the differences in these results.

  1. For the presented bar graphs in Fig. 3 the results of extensive statistical testing should be given. It is not clear, if some of the big differences (cf. Fig. 3D, mean difference more than five fold) are significant or not. 

P-values from t-tests have been added to the histograms in Fig. 3.

  1. The observed higher ATP levels in older neurons are contradicting the lower respiration rates (poorer RC performance) and poorer coupling (lower membrane potential) of mitochondria. This could be a result of composition changes during culture (cf. point 1).

We agree that these data can be interpreted as contradictory. We do not think these represent changes in glial composition (see reply to point 1 above). However, these represent 2 different settings - directly isolated neurons vs. 4 days in vitro- and also represent static and dynamic measurements, respectively.  This is touched on in the discussion (below). The seahorse assay is subject to specific manipulation of the mitochondria and changes in oxidative stress that are not represented in the static ATP measurement. These changes in the cellular environment may have significant impact on the ATP levels and make direct comparison between these two measures challenging.

“…….. the assay used measures ATP present in the cell at the time of lysis rather than dynamic ATP production. Therefore, the absolute ATP numbers may be higher in older mice due to a dysfunctional or slower metabolism of ATP. This could indicate the increased storage but under-utilization of the ATP produced. Alternatively, for various reasons older animals may require increased ATP to perform compared to younger counterparts [69]. It must also be taken into account that this ATP measurement and the seemingly contradictory data from the Seahorse respiratory analysis are produced from two different experiments, under two different conditions. Specifically, this static ATP measure is from cell lysate analysis while the Seahorse measurement are taken from live cells in vitro, and we do not know how this may affect the ATP levels.. …..”

  1. The English requires a very careful editing.

We have now performed a more detailed editing, by authors and by a professional editor.

Round 2

Reviewer 1 Report

The authors have categorically replied to each query and provided much rationale for why their data, in particular the OCR/ECAR data, were extremely variable and not significant. They also agree that the entire study is underpowered. Accordingly, the doubling of the Discussion content with justification for their insignificant findings does not convince this reviewer that the study can be repeated as presented, notably considering the numerous caveats argued. The responses to the queries are quite deflective and, instead, they provide the manufacturer’s kit details without providing a scientific basis for their results. Even more, the biggest issue is normalization among groups for cell numbers at end of Seahorse assays. How/was Seahorse data normalized? As they mentioned in rebuttal, they obtained a limited number of live cortical neurons from each cortex, so did they load equal numbers of live neurons and did they do cell number standardizations for Seahorse?   

Author Response

We thank the reviewer for the helpful comments. Please see the attachment for our point-by-point response.

Best

Cedric Geoffroy

Reviewer 2 Report

I have no further comments. 

Author Response

This reviewer did not have any further comments. Thank you.

Author Response

This reviewer did not have any comments. Thank you.

Reviewer 4 Report

The authors have addressed all of my concerns adequately.

Author Response

This reviewer mentioned we previously replied to all his concerns. Thank you.